# Antenatal care utilisation and receipt of its components in Nigeria: Assessing disparities between rural and urban areas—A nationwide population-based study

Emmanuel O. Adewuyi[1]*, Asa Auta[2], Mary I. Adewuyi[3], Aaron Akpu Philip[4], Victory Olutuase[5], Yun Zhao[6], Vishnu Khanal[7]

1 School of Medical and Health Sciences, Centre for Precision Health, Edith Cowan University, Joondalup, Western Australia, Australia, 2 Faculty of Health, Social Care and Medicine, Edge Hill University, Ormskirk, United Kingdom, 3 Faculty of Health, Department of Social Work, Charles Darwin University, Darwin, Northern Territory, Australia, 4 Research and Development, Australian Red Cross Lifeblood, Brisbane, Queensland, Australia, 5 Department of Clinical Pharmacy and Pharmacy Practice, University of Jos, Jos, Nigeria, 6 School of Population Health, Faculty of Health Science, Curtin University, Perth, Western Australia, Australia, 7 Menzies School of Health Research, Charles Darwin University, Alice Springs, Northern Territory, Australia

* e.adewuyi@ecu.edu.au

## Abstract

### Introduction

Antenatal care (ANC) is crucial for positive pregnancy outcomes, but it is underutilised in Nigeria, suggesting unmet needs, and potentially contributing to the country's high burden of maternal and neonatal mortalities. This study comprehensively assesses ANC utilisation and receipt of its components in Nigeria, focusing on disparities between rural and urban areas.

### Methods

We used the data disaggregation approach to analyse the Nigeria Demographic and Health Survey 2018. We estimated ANC utilisation, assessed the receipt of ANC components, and identified factors associated with eight or more ($\geq$ 8) ANC contacts nationally and across rural and urban residences.

### Results

Nationwide, only 20.3% of women had $\geq$ 8 ANC contacts, with a significant disparity (P < 0.001) between urban (35.5%) and rural (10.4%) areas in Nigeria. The North-East region had the lowest ANC utilisation nationally (3.7%) and in urban areas (3.0%), while the North-West had the lowest in rural areas (2.7%). Nationally, 69% of mothers received iron supplements, 70% had tetanus injections, and 16% received medicines for intestinal parasites, with urban residents having higher proportions across all ANC components. Maternal and husband education, health insurance, and maternal autonomy were associated with increased ANC odds at the national, rural, and urban residences. However, differences

**Data Availability Statement:** Data cannot be shared publicly because ethical restrictions prohibit the public sharing of the DHS data set. Data are

available from the DHS repository through online requests by researchers who meet the criteria for access to confidential data. Researchers seeking access would need to create an account and apply at https://dhsprogram.com/data/new-user-registration.cfm. Access and permission to use the data are granted to everyone upon request. Authors did not have any special privileges that others would not have in gaining access to the data used in this study. Interested researchers can replicate the findings of this study by obtaining the data from DHS and following the approach detailed in the manuscript's method section.

**Funding:** The author(s) received no specific funding for this work.

**Competing interests:** The authors have declared that no competing interests exist.

exist, with all ethnicities having higher ANC odds than the Hausa/Fulanis in urban areas and the Yorubas demonstrating greater odds than other ethnicities in rural settings. Internet use was significant only in the national context, watching television only in urban settings, while maternal working status, wealth, birth type, religion, and radio listenership were significant in rural areas.

## Conclusion

Our study reveals significant disparities in ANC utilisation and components across Nigeria, with rural residents, particularly in northern regions, as well as socioeconomically disadvantaged and teenage mothers facing notable challenges. A multifaceted approach prioritising the interplay of intersectional factors like geography, socioeconomic status, education, religion, ethnicity, and gender dynamics is essential. Key strategies should include targeted interventions to promote educational opportunities, expand health insurance coverage, leverage internet and context-specific media, and foster socioeconomic empowerment, with priority for underserved populations.

## Introduction

Antenatal care (ANC) is a vital component of comprehensive maternal healthcare services provided by appropriately skilled health professionals to pregnant women (and adolescent girls) to optimise health and enhance positive pregnancy outcomes for mothers and newborns [1, 2]. These services encompass a range of healthcare assessments, screening for potential risk factors, vaccinations, and nutritional supplementation (such as folic acid and iron) [3, 4]. Additionally, ANC involves the prevention or management of pregnancy-related or comorbid disorders, along with health promotional activities [1, 2]. By adopting a dual focus on public health and clinical intervention, ANC serves as a crucial platform for providing positive pregnancy experiences with the overarching goal of decreasing maternal and perinatal morbidity and mortality [1]. Notably, women with adequate ANC contacts are more likely to deliver their babies in healthcare facilities [5, 6] with opportunities for better access to comprehensive emergency obstetric care services when needed [3, 4]. This practice creates a continuum of care that addresses the healthcare needs of pregnant women and establishes a foundation for improved maternal and neonatal health outcomes.

In its recent guidelines in 2016, the World Health Organisation (WHO) recommended a holistic approach to ANC that emphasises woman-centred care tailored to individual needs and prioritises pregnant women's physical, emotional, and social well-being [1, 2]. Unlike the basic or focused ANC model, which requires a minimum of four visits, the WHO now recommends eight or more ANC contacts throughout the pregnancy, starting early in the first trimester and comprising one contact in the first trimester, two in the second trimester, and five in the third trimester [1, 7]. This recommendation covers various aspects of healthcare service provision, including routine medical check-ups, counselling on nutrition, exercise, and childbirth preparedness, as well as addressing social determinants of health [1–3]. Social determinants of health are the conditions and resources that affect people's lives and access to empowerment, money, and resources, significantly influencing health outcomes and contributing to health inequities within and between population groups [8–10]. Moreover, the WHO highlights the importance of continuous support from healthcare providers, respectful

communication, and involvement of women in decision-making processes to enhance the overall quality of ANC and contribute to positive pregnancy experiences for expectant mothers and their developing foetuses [1–3]. Despite its well-established importance and the WHO's evidence-based recommendations, ANC remains underutilised in several low-to-middle-income countries, including Nigeria, highlighting the urgent need for concerted efforts in this respect.

Nigeria bears a disproportionately high burden of global maternal and neonatal mortalities, accounting for over 28% of maternal deaths (> 82,000 in 2020) and ranking second in the absolute number of neonatal deaths, with over 270,000 reported in 2019 [11, 12]. These mortalities are substantial and concerning, especially compared to the country's population, which comprises less than 3% of the world's total [11]. Amidst these alarming statistics, the pivotal role of ANC utilisation becomes evident. Optimal ANC utilisation offers a multifaceted approach to addressing critical health challenges of pregnancy through early detection of diseases or risk factors, implementing preventative measures, offering education and counselling, providing nutritional support, and facilitating access to specialised care (when required). Consequently, efforts to enhance optimal ANC services utilisation represent a key strategy in mitigating Nigeria's high maternal and neonatal mortality burden—aligning with global healthcare objectives [13, 14].

Some studies have assessed factors associated with ANC service utilisation in Nigeria [15–25]. However, several available studies are limited by their generalised approach, primarily focusing on national estimates using pooled datasets, which may inadvertently mask differences between and within population groups [16]. There is a growing appreciation of the importance of using high-quality, disaggregated data studies as an evidence-based approach to address access, survival, and equity disparities across socioeconomic and geographic divides [26–28]. This data disaggregation approach aligns with the WHO's framework for monitoring progress towards Universal Health Coverage which supports disaggregating all socioeconomic and demographic strata measures to better assess equity in intervention coverage, among other factors [29]. Importantly, in its newly released 2024 'Operational Framework for Monitoring Social Determinants of Health Equity' [10], the WHO emphasises that data disaggregation is crucial for monitoring social determinants of health as it enables the identification and analysis of health inequities across different population subgroups. For example, by breaking down data by factors such as geographic location (rural-urban), demographic (age, gender), socioeconomic, and ethnicity, it is possible to uncover disparities and target interventions more effectively [10]. This detailed analysis approach supports the goal of the 2030 Agenda for Sustainable Development to 'leave no one behind' by tracking progress across various Sustainable Development Goals (SDGs) and ensuring equitable access to health-promoting resources and services [10].

Specifically, Nigeria's diverse demographics and healthcare infrastructure underscore the need for understanding ANC utilisation across various subpopulations [30]. The country's notable geographic, demographic and socioeconomic disparities can exacerbate existing health inequities and have implications for policymaking and resource allocation efforts to improve maternal healthcare services. Identifying areas where resources and interventions are most needed can, for example, enable tailored strategies to bridge the gap in ANC utilisation and improve maternal and neonatal health outcomes nationwide. Consistent with this premise, our previous study used data disaggregation to gain insight into Nigeria's underutilisation of ANC services, focusing on differences between rural and urban residences [16]. Given the importance of this subject and the new WHO recommendations, we aim to build upon the study by investigating ANC service utilisation and quality (i.e., receipt of ANC components) in Nigeria using the latest nationally representative demographic and health survey data [30].

While our prior study [16] adhered to the focused ANC model, requiring a minimum of four ANC contacts, the current study adopts the new WHO recommendation of eight or more contacts [1, 2, 31], reflecting alignment with the updated global health guidelines. In addition, we expanded our investigation to examine disparities in receiving essential ANC components between rural and urban areas following the recommended data disaggregation approach. These components, including iron supplementation, tetanus vaccinations, medicines for intestinal parasites, blood pressure checks, and various laboratory tests [1, 2], are crucial for promoting maternal and neonatal health. The present study's objectives, thus, are to assess ANC utilisation of eight or more contacts (following the new WHO guidelines) and evaluate its variation across national, rural, and urban settings in Nigeria. Secondly, we aim to examine the receipt of ANC components and their differences across these settings. Additionally, we investigate geographic, demographic, socioeconomic, and healthcare-related factors associated with ANC use and assess how they differ across national, rural, and urban settings.

This study allows us to capture context-specific factors that may not be discernible using a 'one-size-fits-all' method of pooled datasets [27, 32, 33]. Moreover, we prioritise discussing our findings through the lens of social determinants of health and intersectionality. This approach recognises that social identities are interwoven and shaped by interacting and mutually constituting social processes and structures [34, 35]. Accordingly, our study provides a deeper understanding of the intertwined factors contributing to health inequities in ANC use in Nigeria. Further, the study offers comprehensive, evidence-based insights on ANC utilisation and quality that can inform policies to enhance maternal and neonatal healthcare outcomes. Aligning with global maternal-child healthcare initiatives, findings are expected to contribute to realising the SDG 3 targets of reducing maternal and neonatal mortality burdens [14] in Nigeria.

## Methods

### Study setting

Nigeria is Africa's most populous country, with over 220 million people, nearly half (46.48%) residing in rural areas. The country comprises over 374 ethnic groups and languages, organised into 36 states and the Federal Capital Territory, with six geopolitical zones (regions). Further administrative subdivisions include local government areas (LGAs) within states and wards, also known as enumeration areas (EAs), within LGAs. Approximately 63% of Nigerians live in multidimensional poverty [36], highlighting socioeconomic disparity. Healthcare provision involves both public and private sectors, with public health services shared among the three tiers of government: primary (LGAs), secondary (states), and tertiary level of care (Federal). In 2017, a notable shift occurred in ANC practices following the adoption of the 2016 WHO ANC guideline [1, 2], increasing ANC contacts from a minimum of four to eight in Nigeria [31]. Nigeria struggles with poor population health outcomes. However, national statistics can mask the profound disparities across geographic and socioeconomic divides [37]; hence, the rural-urban data disaggregation approach used in the current study.

### Data source

This study relies on data extracted from the Nigerian Demographic and Health Survey (NDHS) 2018, a nationwide survey conducted every five years in Nigeria, starting in 1990 [30]. The survey adheres to a validated methodology and is collaboratively implemented by the Nigerian National Population Commission (NPC) and various development partners, including technical support from the Inner-City Fund (ICF) International. The NDHS provides essential health and demographic indicators that are up-to-date and nationally representative

[30]. The NDHS 2018, the sixth edition in its series, used a two-stage stratified cluster sampling method involving about 42,000 households and 1,400 clusters [30].

The survey covered 40,427 households and 1,389 clusters from 14 August to 29 December 2018. The selection criteria were predetermined, and validated questionnaires were employed to collect data from eligible men, women, and households [30]. Interviewers administered the questionnaires, and 41,821 women between the ages of 15–49 participated, with 16,984 women from urban areas and 24,837 from rural areas. The eligible women showed a high response rate of 99.3%, with 99.2% from urban areas and 99.4% from rural areas. This study involved the analysis of data from a weighted sample of 21,553 mothers (8,440 in urban and 13,113 in rural areas) who provided complete information on ANC contacts for their most recent live childbirths that occurred within the five years preceding the survey [30].

We accessed the NDHS 2018 data for research in June–August 2023. We used the Children Recode (KR) dataset of the NDHS, which was completely anonymised before we accessed them, ensuring that there was no identifiable information about the survey participants. The KR dataset contains individual records for each child born to the interviewed women within the five years before the survey. It provides extensive details regarding the child's pregnancy, postnatal care, immunisation, general health status, and corresponding data for their mothers. We utilise the data available in this dataset to access the mother's information. The NDHS 2018 sampling procedures, settings, questionnaires, and design have been previously released in a detailed report [30]. The Nigerian National Health Research Ethics Committee granted ethics approval for the survey. Survey participants who were 18 years or older provided informed written consent, and consent was obtained from their parents or guardians if they were below 18. We received approved access to use the data, and given our research is based on a secondary data analysis, no further ethical clearance was required to conduct this study. Access to the data used in this study is freely available online (https://dhsprogram.com/data/available-datasets.cfm) after obtaining the necessary approval from the Demographic and Health Survey (DHS) program.

## Variables

**Outcome variable.** This study's primary outcome variable was ANC services utilisation, which we extracted from the NDHS 2018 data [30]. We defined ANC utilisation as having at least eight contacts, aligning with the updated WHO guideline and in line with practice in recent studies [1, 2, 17, 21–23, 25]. In this context, ANC services encompass pregnancy-related care administered to women and adolescent girls by qualified healthcare professionals [1, 2]. The ANC variable extracted from the NDHS 2018 was dichotomised into two categories: less than eight times contacts ($< 8$, suggesting underutilisation, coded as "0") and at least eight times ANC contacts ($\geq 8$, indicating utilisation, coded as "1"). Mothers who responded with 'don't know' or had missing information regarding the question "How many times did you receive antenatal care during this pregnancy?" were excluded from our analyses. As secondary outcomes, we equally investigated essential components recommended by the WHO for all expectant mothers attending ANC clinics and how their provision differs across rural and urban residences in Nigeria. These ANC components include treating intestinal parasites, administering iron tablets or syrup, administering tetanus toxoid vaccine, blood sample collection, monitoring blood pressure, and collecting urine samples [1, 2].

**Explanatory variables.** We adopted Andersen's behavioural model as the conceptual framework for selecting explanatory variables in this study, which is consistent with practice in previous studies [6, 16, 17]. Briefly, Andersen's behavioural model is widely used as a framework in healthcare research, especially in studies examining factors influencing health service

utilisation. In the context of ANC utilisation, the model considers three key components: predisposing factors (e.g., demographics), enabling factors (e.g., access to resources), and need factors (e.g., perceived health needs) [38]. Considering variables captured in the NDHS 2018, we apply this model to assess how a range of factors are associated with a woman's decision to seek ANC services. The model, as used in our study, is depicted in Fig 1. Following the example of previous studies [6, 16], we reviewed and carefully selected a total of twenty-two

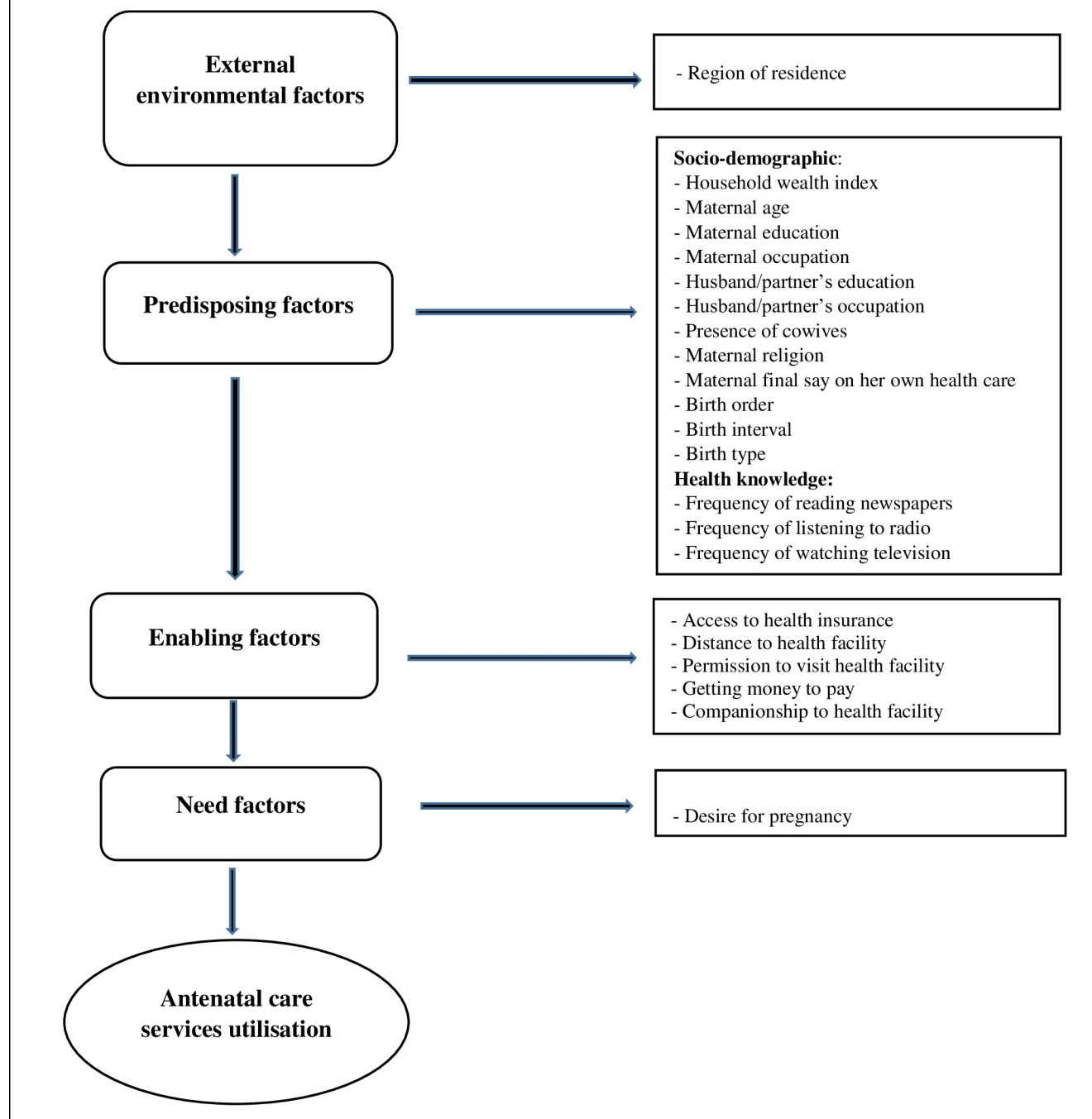

**Fig 1. Theoretical framework to study antenatal care utilisation in Nigeria.**

explanatory variables for inclusion in our analysis. We broadly grouped these variables into four categories: external environmental, predisposing, enabling, and need factors (Fig 1).

External environmental factors include whether the study participants lived in an urban or rural area and which of Nigeria's six geopolitical zones (regions) they reside in: ' North-West, North-East, North-Central, South-South, South-East, and South-West'. We further classified the predisposing variables into health knowledge and socio-demographic factors. Health knowledge factors include media exposure variables such as frequency of reading newspapers/ magazines, internet use, watching television, and listening to the radio. Socio-demographic factors, on the other hand, include the household wealth index, maternal religion, maternal age, maternal and husband/partner's working status, and maternal and husband/partner's education level. Additional socio-demographic factors considered were the mother's final say on her health, ethnicity, preceding birth interval, birth order and birth type. Enabling factors encompass variables that can facilitate healthcare service utilisation, such as access to health insurance, companionship to a health facility, obtaining money to pay for health services, distance to a health facility, and getting permission to visit health facilities. Lastly, the 'desire for pregnancy' was assessed as a need factor. Table 1 provides additional information regarding specific categorisations of these variables.

## Statistical analysis

We assess the unadjusted and adjusted association between ANC utilisation and explanatory variables of interest, first for the overall Nigerian population and then separately for rural and urban residences. We estimated ANC use (proportion of mothers with $\geq 8$ ANC contacts) and its 95% confidence interval (CI) through frequency tabulation and assessed the unadjusted relationship between ANC use and the explanatory variables by using a Chi-Square test. We conducted multivariable binary logistic regression analyses to identify factors significantly associated with ANC utilisation in Nigeria following adjustment for confounders or other predictor variables. In our model-building process, we only included variables that showed statistical significance ($p < 0.05$ in the Chi-Square test) in the initial model of the multivariable regression analyses.

The backward elimination procedure helped to identify significant variables associated with ANC use at a 5% significance level ($p < 0.05$). To minimise potential statistical errors, we meticulously double-checked our analysis process to ensure all variables meeting the inclusion criteria were included in our models. Additionally, we rigorously validated the effectiveness of our backward elimination modelling by testing the final parsimonious models against potential confounding variables and factors previously reported to be associated with ANC. We reported the adjusted odds ratios (AOR), their 95% CI, and p-values for the variables retained in the final model. We excluded missing data or responses recorded as 'don't know' in all analyses. We also estimated the receipt of essential ANC components, first for the overall Nigerian population and thereafter, separately for rural and urban residences.

We used the Statistical Package for Social Sciences (SPSS), version 21 (IBM SPSS Statistics for Windows, Version 21.0. Armonk, NY: IBM Corp) for data management and analyses. In all analyses, we utilised SPSS's complex sample function to adjust for the sample weight and the multistage cluster design of the NDHS 2018 data. This approach addresses the complexities of cluster sampling and stratification, ensuring the statistical reliability of our estimates by accounting for the unequal selection probabilities inherent in the survey design. To facilitate this adjustment, we developed a complex sample plan outlining the necessary considerations for SPSS during data analysis, including sampling weights, strata, and clusters.

**Table 1. Sample characteristics and antenatal care utilisation of ≥ 8 contacts in Nigeria by rural and urban residences.**

| Factors | Nigeria (overall) | | | Urban Nigeria | | | Rural Nigeria | | |
|---|---|---|---|---|---|---|---|---|---|
| | Weighted sample (%)[a] | ANC use (≥8 times ANC contacts) [b] | | Weighted sample (%)[a] | ANC use (≥8 times ANC contacts) [b] | | Weighted sample (%)[a] | ANC use (≥8 times ANC contacts) [b] | |
| | | % (95%CI) | P-Value | | % (95%CI) | P-Value | | % (95%CI) | P-Value |
| *External environmental factors* | | | | | | | | | |
| **Region of residence** | | | <0.001 * | | | <0.001 * | | | <0.001 * |
| North-Central | 3009 (14) | 14.3 (12.4,16.3) | | 939 (11.1) | 25.3 (20.9, 30.4) | | 2071 (15.8) | 9.3 (7.7, 11.0) | |
| North-East | 3858 (17.9) | 3.7 (2.9, 4.6) | | 899 (10.7) | 3.0 (2.1, 4.3) | | 2959 (22.6) | 3.9 (3.0, 5.1) | |
| North-West | 7634 (35.4) | 4.2 (3.5, 5.1) | | 1938 (28) | 8.5 (6.4, 11.2) | | 5697 (43.4) | 2.7 (2.2, 3.4) | |
| South-East | 2104 (9.8) | 38.5 (35.3, 41.8) | | 1531 (18.1) | 37 (33.0, 41.1) | | 573 (4.4) | 42.7 (37.4, 48.1) | |
| South-South | 1932 (9.0) | 39.3 (35.9, 42.9) | | 789 (9.3) | 54.4 (49.2, 59.4) | | 1143 (8.7) | 29.0 (25.1, 33.2) | |
| South-West | 3015 (14) | 63.2 (60.5, 65.8) | | 2344 (27.8) | 67.1 (64.1, 69.9) | | 671 (5.1) | 49.6 (44.0, 55.2) | |
| **Rural-urban residence** | | | <0.001 * | | | | | | |
| Rural | 13113 (60.8) | 10.4 (9.6, 11.3) | | | | | | | |
| Urban | 8440 (39.2) | 35.5 (33.5, 37.6) | | | | | | | |
| *Predisposing factors* | | | | | | | | | |
| **Maternal education level** | | | <0.001 * | | | <0.001 * | | | <0.001 * |
| Higher | 1850 (8.6) | 53.2 (49.7, 56.7) | | 1443 (17.1) | 55.8 (51.5, 59.9) | | 407 (3.1) | 44.1 (39.0, 49.4) | |
| Secondary | 6764 (31.4) | 34.6 (32.9, 36.3) | | 3928 (46.5) | 42.2 (39.8, 44.7) | | 2836 (21.6) | 24.0 (22.0, 26.2) | |
| Primary | 3228 (15) | 19.5 (17.2, 22.1) | | 1298 (15.4) | 29.2 (24.8, 34.2) | | 1930 (14.7) | 13 (11.4, 14.8) | |
| None | 9711 (45.1) | 4.2 (3.7, 4.9) | | 1770 (21.0) | 8.7 (7.0, 10.7) | | 7941 (60.6) | 3.2 (2.7, 3.9) | |
| **Maternal working status** | | | <0.001 * | | | <0.001 * | | | <0.001 * |
| Not working | 6856 (31.8) | 11 (9.9, 12.3) | | 2199 (26.1) | 22.6 (19.8, 25.8) | | 4657 (35.5) | 5.6 (4.8, 6.4) | |
| Working | 14697 (68.2) | 24.6 (23.3, 25.9) | | 6241 (73.9) | 40.0 (37.9, 42.2) | | 8457 (64.5) | 13.1 (12.0, 14.3) | |
| **Husband/partner's education level** | | | <0.001 * | | | <0.001 * | | | <0.001 * |
| Higher | 3053 (15.3) | 38.2 (35.2, 41.4) | | 2051 (26.5) | 46.0 (42.1, 50.0) | | 1002 (8.2) | 22.3 (19.0, 26.0) | |
| Secondary | 6833 (34.2) | 30.2 (28.4, 32) | | 3445 (44.5) | 40.5 (37.7, 43.4) | | 3388 (27.6) | 19.6 (17.8, 21.6) | |
| Primary | 2795 (14) | 17.8 (15.8, 19.9) | | 998 (12.9) | 30.5 (26.7, 34.6) | | 1797 (14.7) | 10.7 (9.0, 12.8) | |
| None | 7322 (36.6) | 4.3 (3.7, 5.1) | | 1243 (16.1) | 11.2 (8.5, 14.6) | | 6079 (49.6) | 2.9 (2.4, 3.6) | |
| **Husband/partner's working status** | | | <0.001 * | | | <0.001 * | | | <0.001 * |

*(Continued)*

**Table 1.** (*Continued*)

| Factors | Nigeria (overall) | | | Urban Nigeria | | | Rural Nigeria | | |
|---|---|---|---|---|---|---|---|---|---|
| | Weighted sample (%)[a] | ANC use (≥8 times ANC contacts) [b] | | Weighted sample (%)[a] | ANC use (≥8 times ANC contacts) [b] | | Weighted sample (%)[a] | ANC use (≥8 times ANC contacts) [b] | |
| | | % (95%CI) | P-Value | | % (95%CI) | P-Value | | % (95%CI) | P-Value |
| Not working | 676 (3.3) | 8.9 (6.4, 12.2) | | 221 (2.8) | 20.0 (13.9, 27.8) | | 455 (3.7) | 3.5 (2.1, 5.8) | |
| Working | 19576 (96.7) | 20.5 (19.3, 21.6) | | 7578 (97.2) | 36.3 (34.2, 38.5) | | 11997 (96.3) | 10.4 (9.6, 11.4) | |
| **Wealth index** | | | <0.001 * | | | <0.001 * | | | <0.001 * |
| Rich | 7625 (35.4) | 40.1 (38.1, 42.3) | | 5685 (67.4) | 43.3 (40.8, 45.9) | | 1941 (14.8) | 30.9 (27.8, 34.2) | |
| Middle | 4396 (20.4) | 17.2 (15.5, 19) | | 1665 (19.7) | 21.5 (18.8, 24.4) | | 2731 (20.8) | 14.6 (12.6, 16.8) | |
| Poor | 9532 (44.2) | 5.8 (5.1, 6.6) | | 1090 (12.9) | 16.2 (12.3, 21.1) | | 8441 (64.4) | 4.4 (3.8, 5.1) | |
| **Maternal age (years)** | | | <0.001 * | | | <0.001 * | | | <0.001 * |
| 15–19 | 1204 (5.6) | 9.5 (7.7, 11.6) | | 239 (2.8) | 23.0 (17.5, 29.6) | | 966 (7.4) | 6.1 (4.5, 8.3) | |
| 20–24 | 4140 (19.2) | 14.4 (13, 15.9) | | 1288 (15.3) | 27.5 (24.2, 31.0) | | 2853 (21.8) | 8.5 (7.3, 9.9) | |
| 25–29 | 5575 (25.9) | 20.2 (18.7, 21.8) | | 2227 (26.4) | 35.8 (32.9, 38.8) | | 3348 (25.5) | 9.8 (8.7, 11.1) | |
| 30–34 | 4704 (21.8) | 25.3 (23.4, 27.3) | | 2149 (25.5) | 40.5 (37.4, 43.8) | | 2555 (19.5) | 12.4 (11.1, 13.9) | |
| 35–39 | 3548 (16.5) | 24.9 (23.1, 26.8) | | 1601 (19.0) | 39.3 (36.1, 42.6) | | 1947 (14.8) | 13.0 (11.3, 15.0) | |
| 40–44 | 1690 (7.8) | 21.3 (18.7, 24.1) | | 666 (7.9) | 36.6 (31.5, 42.0) | | 1025 (7.8) | 11.3 (9.4, 13.5) | |
| 45–49 | 691 (3.2) | 14.2 (11.5, 17.4) | | 271 (3.2) | 16.8 (12.2, 22.7) | | 421 (3.2) | 12.6 (9.5, 16.4) | |
| **Maternal religion** | | | <0.001 * | | | <0.001 * | | | <0.001 * |
| Christianity | 8109 (37.6) | 37.1 (35.3, 38.9) | | 4140 (49.1) | 49.3 (46.8, 51.8) | | 3969 (30.3) | 24.3 (22.3, 26.3) | |
| Traditional/others | 116 (0.5) | 5.1 (1.9, 13.2) | | 36 (0.4) | 12.2 (12.0, 36.0) | | 80 (0.6) | 1.9 (0.6, 6.5) | |
| Islam | 13328 (61.8) | 10.2 (9.1, 11.3) | | 4264 (50.5) | 22.3 (19.7, 25.0) | | 9065 (69.1) | 4.5 (3.8, 5.3) | |
| **Ethnicity** | | | <0.001 * | | | <0.001 * | | | <0.001 * |
| Hausa/Fulani | 9604 (44.6) | 4.4 (3.7, 5.1) | | 2466 (29.2) | 8.6 (6.8, 10.8) | | 7138 (54.4) | 2.9 (2.4, 3.6) | |
| Yoruba | 2611 (12.1) | 61.6 (58.7, 64.4) | | 2074 (24.6) | 63.2 (59.7, 66.6) | | 537 (4.1) | 55.5 (50.8, 60.2) | |
| Igbo | 2698 (12.5) | 44 (41, 47) | | 1959 (23.2) | 44.6 (41.0, 48.4) | | 739 (5.6) | 42.2 (37.4, 47.1) | |
| Others | 6639 (30.8) | 17.3 (16.0, 18.8) | | 1940 (23.0) | 30.9 (27.9, 34.1) | | 4699 (35.8) | 11.7 (10.4, 13.2) | |
| **Place of delivery** | | | <0.001 * | | | <0.001 * | | | <0.001 * |
| Home | 12418 (58.7) | 6.6 (5.9, 7.3) | | 3011 (36. 7) | 15.1 (13.2, 17.3) | | 9408 (72.6) | 3.8 (3.3, 4.5) | |

(*Continued*)

**Table 1.** (Continued)

| Factors | Nigeria (overall) | | | Urban Nigeria | | | Rural Nigeria | | |
|---|---|---|---|---|---|---|---|---|---|
| | Weighted sample (%)[a] | ANC use (≥8 times ANC contacts) [b] | | Weighted sample (%)[a] | ANC use (≥8 times ANC contacts) [b] | | Weighted sample (%)[a] | ANC use (≥8 times ANC contacts) [b] | |
| | | % (95%CI) | P-Value | | % (95%CI) | P-Value | | % (95%CI) | P-Value |
| Public | 5963 (28.2) | 32.4 (30.5, 34.4) | | 3225 (39.3) | 40.2 (37.4, 43.0) | | 2738 (21.1) | 23.2 (21.0, 25.6) | |
| Private | 2786 (13.2) | 53 (50.1, 55.8) | | 1969 (24.0) | 57.9 (54.3, 61.4) | | 8.1 (6.3) | 41.1 (36.8, 45.5) | |
| **Birth order** | | | <0.001 * | | | <0.001 * | | | <0.001 * |
| 1 | 3676 (17.1) | 26.2 (24.3, 28.2) | | 1573 (18.6) | 43.3 (39.8, 46.8) | | 2103 (16.0) | 13.4 (11.9, 15.1) | |
| 2–3 | 7130 (33.1) | 26.3 (24.6, 28) | | 3141 (37.2) | 43.3 (40.6, 46.1) | | 3988 (30.4) | 12.8 (11.5, 15.3) | |
| ≥ 4 | 10747 (49.1) | 14.2 (13.3, 15.3) | | 3725 (44.1) | 25.6 (23.5, 27.9) | | 7022 (53.5) | 8.2 (7.4, 9.1) | |
| **Preceding birth interval** | | | 0.012** | | | 0.084 | | | 0.057 |
| < 24 months | 3656 (20.5) | 17.2 (15.5, 19.1) | | 1387 (20.3) | 31.2 (27.7, 34.9) | | 2269 (20.6) | 8.7 (7.3, 10.3) | |
| ≥ 24 months | 14182 (79.5) | 19.4 (18.3, 20.6) | | 5463 (79.7) | 34.3 (32.1, 36.5) | | 8719 (79.4) | 10.1 (9.3, 11.1) | |
| **Birth type** | | | 0.017** | | | 0.808 | | | <0.001 * |
| Multiple | 422 (2) | 25.6 (21.1, 30.7) | | 174 (2.1) | 36.5 (28.5, 45.4) | | 248 (1.9) | 18.0 (13.3, 23.7) | |
| Single | 21130 (98) | 20.1 (19.1, 21.3) | | 8265 (97.9) | 35.5 (33.4, 37.6) | | 12865 (98.1) | 10.3 (9.5, 11.2) | |
| **Final say on own health** | | | <0.001 * | | | <0.001 * | | | <0.001 * |
| Respondent alone | 1911 (9.4) | 39.7 (36.2, 43.3) | | 1034 (13.2) | 57.7 (52.6, 62.6) | | 878 (7.0) | 18.6 (15.7, 21.8) | |
| Respondent and husband/partner | 6274 (30.9) | 31.5 (29.7, 33.3) | | 3325 (42.5) | 42.4 (39.8, 45.1) | | 2949 (23.6) | 19.1 (17.2, 21.3) | |
| Husband/partner/someone else/other | 12129 (59.7) | 11 (10.1, 12.0) | | 3460 (44.3) | 22.9 (20.5, 25.5) | | 8669 (69.4) | 6.2 (5.5, 7.0) | |
| *Health knowledge factors* | | | | | | | | | |
| **Frequency of reading newspaper/magazine** | | | <0.001 * | | | <0.001 * | | | <0.001 * |
| Not at all | 19095 (88.6) | 17.6 (16.5, 18.7) | | 6756 (80) | 32.7 (30.5, 35.0) | | 12339 (94.1) | 9.3 (8.5, 10.1) | |
| < once a week | 1757 (8.2) | 42 (38.6, 45.4) | | 1199 (14.2) | 48.1 (43.9, 52.4) | | 558 (4.3) | 28.7 (24.5, 33.4) | |
| ≥ Once a week | 701 (3.3) | 39.2 (34.3, 44.4) | | 485 (5.7) | 42.9 (36.5, 49.5) | | 216 (1.6) | 31.0 (24.5, 38.4) | |
| **Frequency of listening to radio** | | | <0.001 * | | | <0.001 * | | | <0.001 * |
| Not at all | 10176 (47.2) | 11.3 (10.2, 12.5) | | 2646 (31.4) | 28.5 (25.1, 32.2) | | 7530 (57.4) | 5.3 (4.6, 6.0) | |
| < once a week | 5171 (24) | 25.7 (23.9, 27.5) | | 2357 (27.9) | 37.5 (34.6, 40.5) | | 2813 (21.5) | 15.8 (14.1, 17.7) | |
| ≥ once a week | 6207 (28.8) | 30.4 (28.4, 32.4) | | 3436 (40.7) | 39.5 (36.6, 42,5) | | 2770 (21.0) | 19.1 (17.0, 21.3) | |
| **Frequency of watching television** | | | <0.001 * | | | <0.001 * | | | <0.001 * |

(Continued)

**Table 1.** (Continued)

| Factors | Nigeria (overall) | | | Urban Nigeria | | | Rural Nigeria | | |
|---|---|---|---|---|---|---|---|---|---|
| | Weighted sample (%)[a] | ANC use (≥8 times ANC contacts) [b] | | Weighted sample (%)[a] | ANC use (≥8 times ANC contacts) [b] | | Weighted sample (%)[a] | ANC use (≥8 times ANC contacts) [b] | |
| | | % (95%CI) | P-Value | | % (95%CI) | P-Value | | % (95%CI) | P-Value |
| Not at all | 12085 (56.1) | 8.5 (7.7, 9.4) | | 2537 (30.1) | 18.7 (16.0, 21.7) | | 9548 (72.8) | 5.8 (5.2, 6.5) | |
| < once a week | 3711 (17.2) | 27.6 (24.9, 30.6) | | 1909 (22.6) | 35.4 (30.9, 40.2) | | 1802 (13.7) | 19.4 (17.0, 22.0) | |
| ≥ Once a week | 5757 (26.7) | 40.2 (37.9, 42.6) | | 3993 (47.3) | 46.2 (43.3, 49.2) | | 1763 (13.4) | 26.7 (23.8, 29.7) | |
| **Frequency of Internet use** | | | <0.001 * | | | <0.001 * | | | <0.001 * |
| Not at all | 19531 (90.6) | 16.7 (15.7, 17.7) | | 6768 (80.2) | 30.2 (28.2, 32.3) | | 12763 (97.3) | 9.5 (8.7, 10.4) | |
| < once a week | 327 (1.5) | 44.8 (38.3, 51.6) | | 255 (3.0) | 48.2 (40.4, 56.0) | | 72 (0.6) | 33.1 (23.1, 45.0) | |
| ≥ Once a week | 1695 (7.9) | 56.8 (53, 60.6) | | 1417 (16.8) | 58.7 (54.3, 62.9) | | 278 (2.1) | 47.6 (40.9, 54.4) | |
| *Enabling factors* | | | | | | | | | |
| **Access to health insurance** | | | <0.001 * | | | <0.001 * | | | <0.001 * |
| No | 21083 (97.8) | 19.7 (18.7, 20.8) | | 8117 (96.2) | 34.9 (32.9, 37.0) | | 12966 (98.9) | 10.2 (9.4, 11.1) | |
| Yes | 470 (2.2) | 43.7 (37.4, 50.3) | | 322 (3.8) | 50.7 (44.0, 57.4) | | 147 (1.1) | 28.4 (19.0, 40.2) | |
| **Distance to health facility** | | | <0.001 * | | | 0.095 | | | <0.001 * |
| Big problem | 6064 (28.1) | 15.9 (14.1, 17.7) | | 1477 (17.5) | 39.2 (34.4, 44,3) | | 4587 (35.0) | 8.3 (7.2, 9.7) | |
| Not a big problem | 15489 (71.9) | 22 (20.7, 23.3) | | 6962 (82.5) | 34.7 (32.5, 37.0) | | 8527 (65.0) | 11.6 (10.6, 12.7) | |
| **Permission to visit health facility** | | | 0.004* | | | 0.122 | | | 0.005 * |
| Big problem | 2575 (11.9) | 16.4 (14, 19.2) | | 687 (8.1) | 40.1 (34.1, 46.4) | | 1888 (14.4) | 7.8 (6.2, 9.8) | |
| Not a big problem | 18978 (88.1) | 20.8 (19.6, 22) | | 7753 (91.9) | 35.1 (33.0, 37.3) | | 11225 (85.6) | 10.9 (10.0, 11.9) | |
| **Getting money for health services** | | | <0.001 * | | | <0.001 * | | | <0.001 * |
| Big problem | 10451 (48.5) | 16.1 (14.9, 17.5) | | 3280 (38.9) | 31.6 (28.7, 34.6) | | 7170 (54.7) | 9.1 (8.1, 10.1) | |
| Not a big problem | 11102 (51.5) | 24.1 (22.7, 25.6) | | 5159 (61.1) | 38.0 (35.6, 40.4) | | 5943 (45.3) | 12.1 (10.9, 13.4) | |
| *Need factor* | | | | | | | | | |
| **Desire for pregnancy** | | | <0.001 * | | | 0.335 | | | <0.001 * |
| Then | 18965 (88) | 19.2 (18.1, 20.4) | | 7175 (85) | 35.0 (32.8, 37.3) | | 11790 (89.9) | 9.6 (8.7, 10.5) | |
| Later | 1874 (8.7) | 28 (25.4, 30.8) | | 918 (10.9) | 38 (33.7, 42.6) | | 956 (7.3) | 18.4 (15.7, 21.5) | |

(*Continued*)

**Table 1.** (Continued)

| Factors | Nigeria (overall) | | | Urban Nigeria | | | Rural Nigeria | | |
|---|---|---|---|---|---|---|---|---|---|
| | Weighted sample (%)[a] | ANC use (≥8 times ANC contacts) [b] | | Weighted sample (%)[a] | ANC use (≥8 times ANC contacts) [b] | | Weighted sample (%)[a] | ANC use (≥8 times ANC contacts) [b] | |
| | | % (95%CI) | P-Value | | % (95%CI) | P-Value | | % (95%CI) | P-Value |
| No more | 714 (3.3) | 27.9 (24.2, 32.1) | | 347 (4.1) | 38.3 (31.6, 45.4) | | 367 (2.8) | 18.2 (14.6, 22.4) | |

ANC: Antenatal care, ≥ 8: eight or more ANC contacts (ANC use), CI: confidence interval

[a] weighted sample size and percentages

[b] weighted percentage of ANC use

*Significant at 1% level

**Significant at 5% level

## Results

### Sample characteristics for the overall, rural, and urban residences

Table 1 details the characteristics of the sample examined in our study, including data from rural and urban residences in Nigeria. Our study included a total weighted sample of 21,553 mothers aged 15–49 with information about their ANC contacts. Of these, 60.8% (13,113) were from rural residences, and 39.2% (8,440) were from urban areas. In Nigeria, 5.6% of participants were teenagers. The proportion of teenage mothers was lower in urban areas (2.8%) compared to rural areas (7.4%). A high proportion of participants (45.1%) had no formal education, with this trend being more pronounced in rural areas (60.6%) than in urban areas (21.0%). Similarly, most participants were from poor households (44.2%), with rural areas having a substantially higher percentage (64.4%) compared to urban areas (12.9%). Approximately 60% of mothers delivered their babies at home in Nigeria—36.7% in urban residences and over 70% in rural areas. Internet use at least once a week was low at 7.9%, with rural areas having considerably low usage (2.1%) compared to urban areas (16.8%). Access to health insurance coverage was also low at 2.2% in Nigeria—3.8% in urban and 1.1% in rural areas. Nearly half of women (48.5%) reported having a big challenge with getting money for health services. This challenge was more prominent in rural residences (54.7%) than urban residences (38.9%).

### Eight or more ANC utilisation in Nigeria by rural and urban residences

A total of 4,366 mothers (20.3%, 95% CI: 19.2, 21.4%, P < 0.001) reported having at least eight ANC contacts in Nigeria (Table 1). Urban mothers notably recorded higher ANC utilisation (35.5%, 95% CI: 33.5, 37.6%) than their rural counterparts (10.4%, 95% CI: 9.6, 11.3%, P < 0.001, Fig 2). At the regional level, the South-West geopolitical zone had the highest ANC utilisation at 63.2% (95% CI: 60.5, 65.8%, P < 0.001) in Nigeria, consistent across rural (49.6%, 95% CI: 44.0, 55.2%, P < 0.001) and urban (67.1%, 95% CI: 64.1, 69.9%, P < 0.001) residences (Fig 3). In contrast, the North-East region recorded the lowest ANC use in Nigeria (3.7%, 95% CI: 2.9, 4.6%, P < 0.001), and urban Nigeria (3.0%, 95% CI: 2.1, 4.3%, P < 0.001), while the North-West region had the lowest in rural Nigeria (2.7%, 95% CI: 2.2, 3.4%, P < 0.001) as summarised in Fig 3.

Considering predisposing factors, all socio-demographic and health knowledge variables were significantly associated (to varying degrees) with ANC utilisation in Nigeria, rural or urban residences notwithstanding (Table 1). However, there were two exceptions, namely

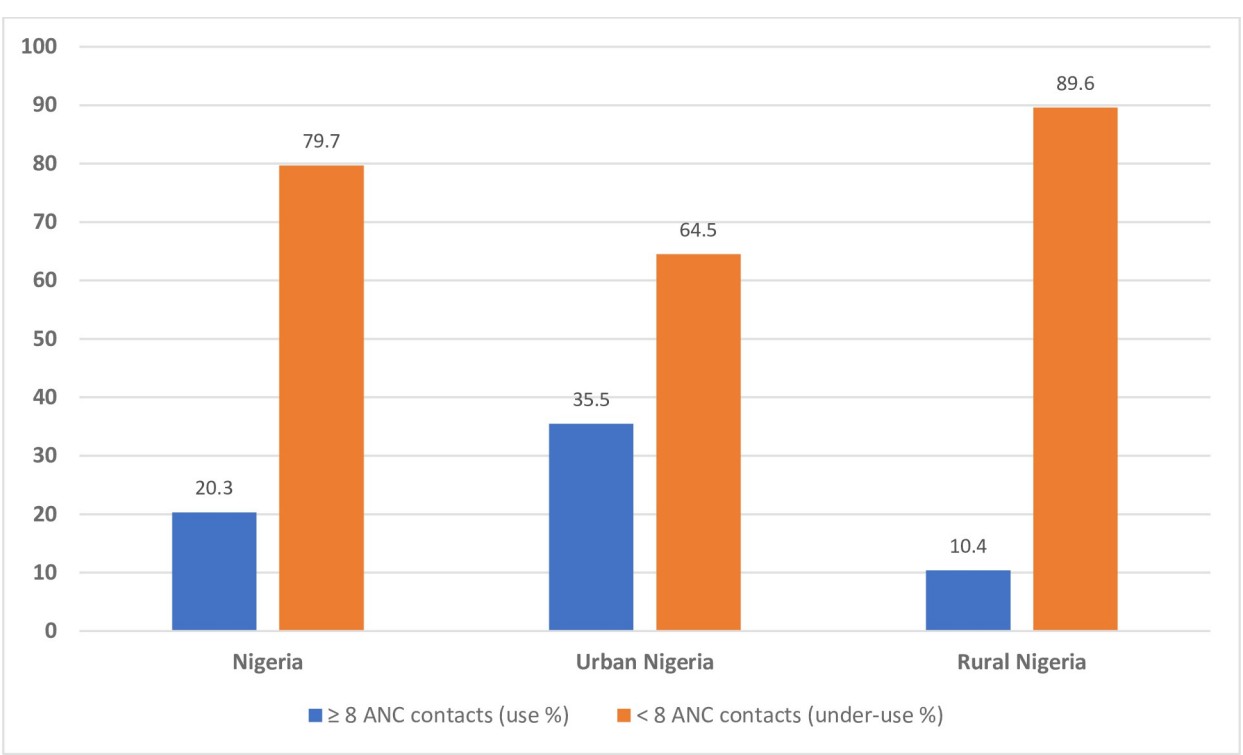

**Fig 2. Antenatal care use and under-use in Nigeria by rural-urban residences.**

'preceding birth interval' (significantly associated with ANC use only in the overall population) and 'birth type', which did not reach statistical significance in urban areas (Table 1). Among the enabling factors, distance and permission to visit health facilities were not statistically significant in urban areas. The 'need factor' also showed a similar relationship with ANC utilisation, where the 'desire for pregnancy' was significant in the overall population and rural residences but not in urban areas. Table 1 provides additional details on ANC use for all the predictor variables across Nigeria's overall, rural, and urban residences.

## ANC utilisation based on the focused model (four or more ANC contacts)

Although the focused ANC model is no longer recommended in Nigeria, we present these results to facilitate a comparison of ANC use (and underuse) between the current study (using the NDHS 2018 data) and our previous study utilising the NDHS 2013 data [16]. We first summarise the results based on the NDHS 2018, in which a total of 12,456 (57.8%) mothers had four or more (≥4) ANC contacts in Nigeria—6,418 (76.1%) in urban areas and 6,038 (46.0%) in rural areas. Conversely, 9,097 (42.2%) had less than four ANC contacts (underuse) in Nigeria—2,021 (23.9%) in urban and 7,075 (54%) in rural settings. Notably, 5,336 mothers (24.4%) did not have ANC contact at all in the overall Nigerian population—878 (10.1%) were in urban residences, and 4,458 (33.8%) were in rural residences. Additionally, 574 (2.6%) mothers had only one ANC contact: 158 [1.8%] in urban and 416 [3.2%] in rural areas, while 970 (4.4%) had two contacts: 264 [3.0%] in urban and 706 [5.4%] in rural areas. Finally, 2,216 (10.1%) mothers had three contacts: 722 [8.3%] in urban and 1494 [11.3%] in rural areas in Nigeria.

For comparison, in 2013 [16], only 53.5% of mothers reported having ≥4 ANC contacts. This percentage was higher in urban areas at 77.6% compared to 38.9% in rural areas [16].

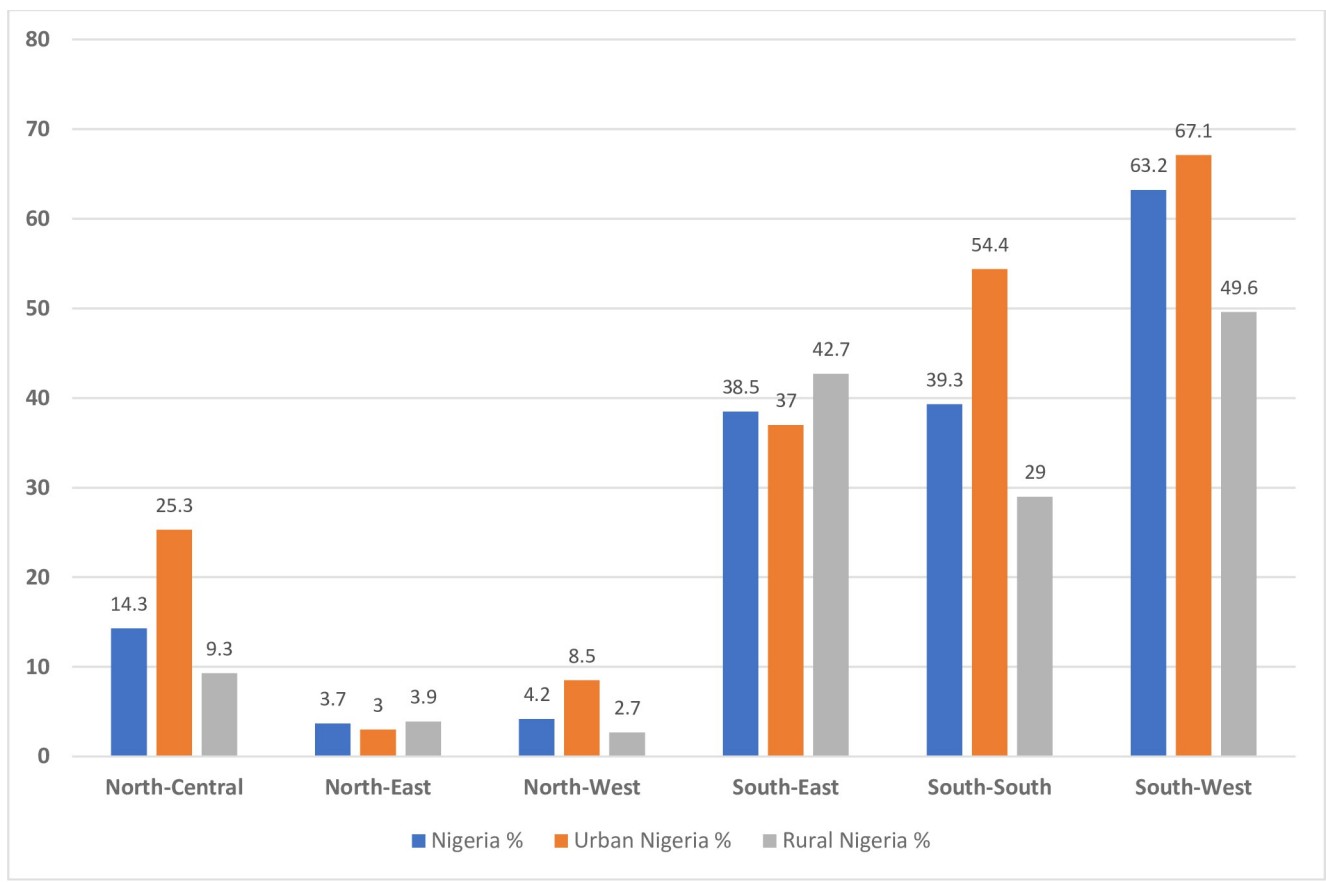

**Fig 3. Antenatal care utilisation in Nigeria's geopolitical zones by rural-urban residences.**

There was significant underuse of ANC services, with 1.8% having one contact, 3.8% having two contacts, 6.8% having three contacts, and 33.9% having no contacts at all [16]. An estimated 46.5% of mothers did not meet the recommended ≥4 ANC contacts, with much higher underuse among rural women at 61.1% compared to urban women at 22.4% [16]. Comparing the 2013 findings with the current results, there was an improvement in ≥4 ANC use between 2013 and 2018, particularly in the national and rural settings, with the overall percentage increasing from 53.5% in 2013 to 57.8% in 2018 and the rural percentage rising from 38.9% in 2013 to 46.0% in 2018. However, this positive trend did not extend to urban areas, where ANC use decreased from 77.6% in 2013 to 76.1% in 2018 (i.e., underuse slightly increased from 22.4% in 2013 to 23.9% in 2018). Although there was a decrease in the percentage of mothers with no ANC contacts over the five years, dropping from 33.9% in 2013 to 24.4% in 2018, the rural-urban disparity remained. In 2013, 77.6% of urban mothers had ≥4 contacts compared to only 38.9% of rural mothers. In 2018, the figures improved (for rural residences), with 76.1% of urban mothers and 46.0% of rural mothers meeting the ≥4 ANC contacts, but a significant rural-urban gap persisted.

## Components of ANC received in Nigeria across rural-urban settings

Table 2 presents a comprehensive breakdown of ANC components in Nigeria, including the provision of iron tablets or syrup, tetanus injections, drugs for intestinal parasites, blood pressure, urine samples, and blood sample assessments during pregnancy, with a focus on the

**Table 2. Components of ANC received in Nigeria disaggregated by rural and urban residences.**

| S/N | Factors | Overall (Nigeria) | | | Urban Nigeria | | | Rural Nigeria | | |
|---|---|---|---|---|---|---|---|---|---|---|
| | | N[a] | %[b] (95%CI) | P-value[c] | N[a] | %[b] (95%CI) | P-value[c] | N[a] | %[b] (95%CI) | P-value[c] |
| 1 | Given or bought iron tablets/syrup | | | | | | | | | |
| | Yes | 15189 | 69.3 (67.7, 70.9) | < 0.001* | 6978 | 80.1 (77.8, 82.2) | < 0.001* | 8211 | 62.2 (60.0, 64.4) | < 0.001* |
| 2 | Received tetanus injections before birth | | | | | | | | | |
| | Yes | 15267 | 70.0 (68.4, 71.5) | < 0.001* | 7290 | 84.3 (82.6, 85.9) | < 0.001* | 7977 | 60.6 (58.4, 62.8) | < 0.001* |
| 3 | Had drugs for intestinal parasites during pregnancy | | | | | | | | | |
| | Yes | 3666 | 16.7 (15.9, 17.6) | < 0.001* | 1624 | 18.6 (17.4, 19.9) | < 0.001* | 2042 | 15.5 (14.4, 16.7) | < 0.001* |
| 4 | During pregnancy: blood pressure taken | | | | | | | | | |
| | Yes | 15566 | 93.9 (93.2, 94.5) | < 0.001* | 7525 | 96.1 (95.1, 96.9) | < 0.001* | 8042 | 92.0 (91.0, 92.9) | < 0.001* |
| 5 | During pregnancy: urine sample taken | | | | | | | | | |
| | Yes | 14313 | 86.4 (85.2, 87.4) | < 0.001* | 7148 | 91.2 (89.8, 92.5) | < 0.001* | 7165 | 82.0 (80.3, 83.5) | < 0.001* |
| 6 | During pregnancy: blood sample taken | | | | | | | | | |
| | Yes | 14522 | 87.6 (86.6, 88.6) | < 0.001* | 7215 | 92.1 (90.6, 93.3) | < 0.001* | 7307 | 83.6 (82.1, 85.0) | < 0.001* |

ANC: antenatal care N[a]: weighted sample, %[b]: weighted percentage, S/N: serial number, [c]P-value for the analysis assessing yes vs no
*Significant at 1% level.

rural-urban divide. In the overall Nigerian context, 69.3% received iron tablets or syrup during pregnancy, 70.0% had tetanus injections, and 16.7% received drugs for intestinal parasites. Blood pressure, urine samples, and blood sample assessments were conducted for 93.9%, 86.4%, and 87.6% of pregnant women, respectively. Examining urban and rural disparities reveals that urban residents generally exhibit higher percentages across all components of ANC, with considerable differences observed in the provision of iron tablets or syrups (80.1% urban vs. 62.2% rural) and tetanus injections (84.3% urban vs. 60.6% rural). The gap is narrower for other components but consistently favours urban settings (Table 2).

## Factors associated with eight or more ANC contacts in Nigeria by urban and rural residences

Following adjustment for other predictors in the multivariable analyses, several factors were independently associated with ANC utilisation (of $\geq$ 8 contacts) in the overall Nigerian population (Table 3). These factors include the region of residence, rural-urban residences, maternal educational level, husband educational level, health insurance coverage, maternal age, maternal working status, wealth index, desire for pregnancy, birth order, maternal healthcare decision-making, maternal religion, and ethnicity (Table 3). The frequency of internet use was only marginally significant in the overall Nigerian population.

In urban residences, all geopolitical zones had significantly higher odds of ANC utilisation than the North-East region, with the South-West (OR = 31.09, 95%CI: 18.71–51.67, P < 0.001) and South-South (OR = 22.51, 95%CI: 13.83–36.63, P < 0.001) having substantially greater odds than others (Table 3). Mothers with at least a secondary level education (particularly those with higher education) had increased odds of ANC utilisation compared to their uneducated counterparts. This finding was slightly different for 'husband's education level', where only the higher level of attainment increased the odds of ANC use. Access to health insurance significantly increased the odds of ANC use. Moreover, the odds of ANC use were noticeably higher among mothers aged 25–44 years. Compared to the birth order of four or more, the odds of ANC were higher for lower birth orders. Mothers who could decide independently on healthcare matters had greater odds of ANC use. Notably, the odds of using ANC were greater

**Table 3. Factors associated with ANC utilisation of $\geq$ 8 contacts in Nigeria across rural and urban residences.**

| Factors | Overall (Nigeria) | | | Urban Nigeria | | | Rural Nigeria | | |
|---|---|---|---|---|---|---|---|---|---|
| | AOR | 95%CI | P | AOR | 95%CI | P | AOR | 95%CI | P |
| **Region** | | | < 0.001* | | | < 0.001* | | | < 0.001* |
| North-Central | 2.42 | 1.79, 3.27 | < 0.001* | 7.44 | 4.64, 11.94 | < 0.001* | 1.29 | 0.87, 1.93 | 0.208 |
| North-West | 1.15 | 0.82, 1.63 | 0.416 | 3.39 | 2.00, 5.75 | < 0.001* | 0.68 | 0.44, 1.04 | 0.077 |
| South-East | 3.32 | 2.18, 5.07 | < 0.001* | 7.05 | 3.96, 12.57 | < 0.001* | 3.04 | 1.64, 5.64 | <0.001* |
| South-South | 6.73 | 4.80, 9.44 | < 0.001* | 22.51 | 13.83, 36.63 | < 0.001* | 3.12 | 1.95, 4.99 | <0.001* |
| South-West | 10.93 | 7.66, 15.60 | < 0.001* | 31.09 | 18.71, 51.67 | < 0.001* | 5.05 | 2.86, 8.91 | <0.001* |
| North-East (Ref) | 1.00 | Reference | — | 1.00 | Reference | — | 1.00 | Reference | — |
| **Urban-rural residences** | | | < 0.001* | | | | | | |
| Urban | 1.33 | 1.17, 1.52 | < 0.001* | | | | | | |
| Rural | 1.00 | Reference | — | | | | | | |
| **Maternal education level** | | | < 0.001* | | | < 0.001* | | | < 0.001* |
| Primary | 1.40 | 1.13, 1.73 | 0.011** | 1.31 | 0.94, 1.84 | 0.111 | 1.47 | 1.13, 1.92 | 0.004* |
| Secondary | 1.69 | 1.37, 2.09 | < 0.001* | 1.65 | 1.19, 2.29 | 0.003* | 1.73 | 1.32, 2.27 | <0.001* |
| Higher | 2.16 | 1.64, 2.84 | < 0.001* | 2.28 | 1.53, 3.38 | < 0.001* | 2.18 | 1.53, 3.11 | <0.001* |
| No education | 1.00 | Reference | — | 1.00 | Reference | — | 1.00 | Reference | — |
| **Husband education level** | | | < 0.001* | | | 0.003* | | | 0.013** |
| Primary | 1.17 | 0.91, 1.50 | 0.219 | 1.11 | 0.73, 1.68 | 0.633 | 1.17 | 0.87, 1.57 | 0.291 |
| Secondary | 1.42 | 1.12, 1.80 | 0.004* | 1.34 | 0.89, 2.01 | 0.159 | 1.44 | 1.09, 1.89 | 0.01** |
| Higher | 1.85 | 1.39, 2.45 | < 0.001* | 1.85 | 1.15, 2.97 | 0.011** | 1.68 | 1.21, 2.34 | 0.001* |
| No education | 1.00 | Reference | — | 1.00 | Reference | — | 1.00 | Reference | — |
| **Health insurance coverage** | | | 0.001* | | | 0.043** | | | 0.014** |
| Yes | 1.64 | 1.22, 2.22 | 0.001* | 1.44 | 1.01, 2.05 | 0.043** | 2.09 | 1.16, 3.74 | 0.014** |
| No | 1.00 | Reference | — | 1.00 | Reference | — | 1.00 | Reference | — |
| **Maternal healthcare decision** | | | < 0.001* | | | < 0.001* | | | 0.029** |
| Respondent alone | 1.64 | 1.32, 2.02 | < 0.001* | 1.83 | 1.40, 2.41 | < 0.001* | 1.53[b] | 1.11, 2.12 | 0.01** |
| Respondent and spouse | 0.88 | 0.76, 1.02 | 0.081 | 0.874 | 0.72, 1.06 | 0.178 | 1.11[a] | 0.90, 1.36 | 0.332 |
| Spouse alone/someone else/others | 1.00 | Reference | — | 1.00 | Reference | — | 1.00[c] | Reference | — |
| **Maternal age** | | | 0.007* | | | 0.009* | | | 0.003* |
| 20–24 | 1.03 | 0.74, 1.43 | 0.85 | 1.21 | 0.70, 1.10 | 0.703 | 1.16 | 0.74, 1.82 | 0.515 |
| 25–29 | 1.17 | 0.84, 1.65 | 0.36 | 1.64 | 1.01, 2.65 | 0.048** | 1.11 | 0.71, 1.75 | 0.649 |
| 30–34 | 1.40 | 1.00, 1.97 | 0.05** | 1.75 | 1.06, 2.88 | 0.028** | 1.60 | 1.02, 2.53 | 0.042** |
| 35–39 | 1.46 | 1.02, 2.09 | 0.04** | 1.88 | 1.16, 3.03 | 0.01** | 1.53 | 0.95, 2.47 | 0.081 |
| 40–44 | 1.64 | 1.10, 2.43 | 0.015** | 2.15 | 1.27, 3.62 | 0.004* | 1.61 | 0.95, 2.73 | 0.075 |
| 45–49 | 1.31 | 0.81, 2.12 | 0.267 | 1.19[d] | 0.57, 2.49 | 0.572 | 2.09 | 1.14, 3.84 | 0.017** |
| 15–19 | 1.00 | Reference | — | 1.00[e] | Reference | — | 1.00 | Reference | — |
| **Maternal working status** | | | 0.007* | | | | | | 0.011** |
| Yes | 1.20 | 1.05, 1.37 | 0.007* | | | | 1.29 | 1.06, 1.56 | 0.011** |
| No | 1.00 | Reference | — | | | | 1.00 | Reference | — |
| **Wealth index** | | | < 0.001* | | | | | | < 0.001* |
| Rich | 1.47 | 1.22, 1.76 | 0.003* | | | | 1.55 | 1.22, 1.98 | <0.001* |
| Middle | 1.29 | 1.09, 1.53 | < 0.001* | | | | 1.45 | 1.19, 1.76 | <0.001* |
| Poor | 1.00 | Reference | — | | | | 1.00 | Reference | — |
| **Desire for pregnancy** | | | 0.01 | | | | | | |
| Then vs. No more | 1.43 | 1.07, 1.92 | 0.017 | | | | | | |
| Later vs. No more | 1.24 | 0.87, 1.75 | 0.237 | | | | | | |
| No more | 1.00 | Reference | — | | | | | | |

(*Continued*)

**Table 3.** (Continued)

| Factors | Overall (Nigeria) | | | Urban Nigeria | | | Rural Nigeria | | |
|---|---|---|---|---|---|---|---|---|---|
| | AOR | 95%CI | P | AOR | 95%CI | P | AOR | 95%CI | P |
| **Birth order** | | | < 0.001* | | | 0.001* | | | 0.001* |
| 1 | 1.45 | 1.22, 1.74 | < 0.001* | 1.44 | 1.14, 1.82 | 0.002* | 1.67 | 1.28, 2.18 | <0.001* |
| 2nd - 3rd | 1.28 | 1.13, 1.45 | < 0.001* | 1.349 | 1.15, 1.59 | < 0.001* | 1.30 | 1.08, 1.56 | 0.006* |
| ≥ 4 | 1.00 | Reference | — | 1.00 | Reference | — | 1.00 | Reference | — |
| **Maternal religion** | | | 0.006* | | | | | | 0.002* |
| Christianity | 5.95 | 1.95, 18.17 | 0.002* | | | | 9.50 | 2.77, 32.61 | < 0.001* |
| Islam | 5.42 | 1.75, 16.74 | 0.003* | | | | 8.47 | 2.38, 30.14 | 0.001* |
| Traditionalist/others | 1.00 | Reference | — | | | | 1.00 | Reference | — |
| **Ethnicity** | | | < 0.001* | | | < 0.001* | | | 0.002* |
| Yoruba | 1.93 | 1.32, 2.81 | < 0.001* | 2.07 | 1.31, 3.25 | 0.002* | 2.40 | 1.22, 4.73 | 0.011** |
| Igbo | 2.37 | 1.63, 3.45 | < 0.001* | 3.16 | 2.06, 4.85 | < 0.001* | 1.79 | 0.97, 3.31 | 0.062 |
| Others | 1.16 | 0.86, 1.58 | 0.33 | 1.53 | 1.05, 2.21 | 0.026** | 1.04 | 0.66, 1.65 | 0.870 |
| Hausa/Fulani | 1.00 | Reference | — | 1.00 | Reference | — | 1.00 | Reference | — |
| **Frequency of Internet use** | | | 0.049** | | | | | | |
| < Once a week | 0.84 | 0.60, 1.18 | 0.315 | | | | | | |
| ≥ Once a week | 1.22 | 1.003, 1.48 | 0.047** | | | | | | |
| Not at all | 1.00 | Reference | — | | | | | | |
| **Frequency of watching TV** | | | | | | 0.048** | | | |
| Not at all | | | | 1.32 | 1.04, 1.67 | 0.022** | | | |
| ≥ Once a week | | | | 1.29 | 0.99, 1.68 | 0.062 | | | |
| < Once a week | | | | 1.00 | Reference | — | | | |
| **Frequency of listening to radio** | | | | | | | | | <0.001* |
| < once a week vs. Not at all | | | | | | | 1.51 | 1.23, 1.86 | <0.001* |
| ≥ once a week vs. Not at all | | | | | | | 1.69 | 1.37, 2.09 | <0.001* |
| Not at all | | | | | | | 1.00 | Reference | — |
| **Birth type** | | | | | | | | | 0.007* |
| Multiple | | | | | | | 1.74 | 1.16, 2.61 | 0.007* |
| Single | | | | | | | 1.00 | Reference | — |

AOR: Adjusted odds ratio, CI: confidence interval

* Significant at 1% level

** Significant at 5% level, for 'maternal healthcare decision' variable in rural area

'a' which on the Table corresponds to 'respondent and spouse' was changed to 'spouse alone/someone else/others'

'b' to 'respondent alone'

'c' to 'respondent and spouse' (i.e., the reference category for this variable was changed to the 'respondent and spouse' so we can make a meaningful comparison in this instance in rural residences), for the 'maternal age' variable in urban area

'd' which on the Table corresponds to 45–49 age category was changed to 15–19, and

'e' to 45–49 category (i.e., the reference category in this case was '45–49', which enables meaningful comparison in urban residences). We analysed factors associated with ANC utilisation in Nigeria using multivariable binary logistic regression. We only included variables with statistical significance (p < 0.05) and identified significant factors using backward elimination at a 5% significance level (p < 0.05). We reported AOR, 95% CI, and p-values for the variables retained in the final model. We used the same analysis procedure for rural and urban populations and considered potential confounders. The significant factors may differ across the residences, and we leave the space vacant where a variable is not significant.

for all ethnicities compared to the Hausa/Fulani. Watching television at least once a week similarly increased the odds of ANC use in urban Nigeria.

In rural residences, the odds of ANC utilisation were significantly higher only in the southern geopolitical zones of South-West, South-East, and South-South (Table 3). All the other

regions (northern geopolitical zones) had no statistically significant difference in their odds of ANC use compared to the North-East. Acquiring at least a primary level of education significantly increased the odds of ANC use, with the 'higher level' having the greatest odds. Mothers whose husbands acquired at least a primary education level had increased odds of ANC utilisation with greater odds for 'higher education' compared to the 'no education' category. The odds of ANC use similarly increased for mothers with access to health insurance coverage compared to those without such facility. The odds of ANC use were only significantly higher among mothers aged 30–34 and those aged 45–49. Mothers who engaged in paid jobs had greater odds of ANC use than those who were not. Belonging to middle or wealthy households increased the odds of ANC use. Compared to the birth order of four or more, lower birth orders were associated with increased odds of ANC utilisation. Mothers who could decide on their own concerning their health had greater odds of ANC use. Mothers who identified as Christians or Muslims had increased odds of ANC use compared to those who identified as Traditional or other religions. Christian mothers had slightly higher odds than their Muslim counterparts. Compared to the Hausa/Fulani ethnic group, only the Yoruba had significantly increased odds of ANC use. Mothers who listened to the radio (compared to their counterparts who did not) and those who gave birth to multiple foetuses (compared to singletons) had increased odds of ANC use.

## Comparing factors associated with eight or more ANC contacts in rural and urban residences

In urban and rural residences across Nigeria, ANC utilisation shares some common factors, particularly the influence of education, access to health insurance, and maternal autonomy. Educational attainment consistently emerges as a significant factor in both settings. Mothers with at least a secondary education, particularly those with higher education, demonstrated increased odds of ANC use in rural and urban areas. Access to health insurance coverage also increased the odds of ANC use in urban and rural settings. Maternal autonomy (making independent decisions regarding healthcare matters) is a shared factor positively associated with ANC utilisation in both settings. Despite these commonalities, differences exist between urban and rural residences.

In urban areas, geographical disparities were prominent, with all the regions having significantly higher odds of ANC use than the North-East. In rural areas, the regional disparity follows a different pattern, with higher odds of ANC observed only in the southern regions—all the northern regions had substantially lower odds of ANC use than their southern counterparts, with the North-West performing worse than other northern regions. In urban areas, secondary and higher maternal education levels increased the odds of ANC use. In rural areas, primary, secondary, and higher education levels increased the odds of ANC use. In urban residences, only a 'higher husband's education' level increased the odds of ANC use, while secondary and higher husband's education levels were significant in rural settings. Maternal age between 25–44 years increased the odds of ANC use in urban residences, while specific age ranges of 30–34 and 40–45 demonstrate increased odds in rural areas. In urban areas, all ethnic groups had higher odds of ANC use than the Hausa/Fulani. In rural areas, only the Yorubas had greater ANC odds. The frequency of watching television was significant only in urban areas. Conversely, maternal working status, wealth index, birth type, maternal religion, and frequency of listening to radio attained statistical significance only in rural residences.

## Discussion

We present a comprehensive assessment of ANC utilisation, components, and factors associated with ≥8 ANC contacts across Nigeria's national, rural, and urban settings. The

proportion of mothers achieving the recommended $\geq 8$ ANC contacts was 20.3%, with urban residents showing about 1.70-fold higher utilisation (35.5%) and rural mothers demonstrating about 2-fold lower use (10.4%). We previously investigated the rural-urban differences in ANC underuse based on the focused model and the NDHS 2013 [16]. In the current study, 42.2% of mothers recorded < 4 ANC contacts (underuse, based on the old model) in Nigeria —54% in rural and 23.9% in urban settings. Compared to the 2013 data, these findings (for the old model) suggest a marginal decrease in ANC underuse in Nigeria (46.5% in 2013 [16]) and rural residences (61.1% in 2013 [16]) but not in urban areas (22.4% in 2013 [16])—indicating an improvement. However, considering the new recommendation for $\geq 8$ ANC contacts, there was no substantial improvement in ANC use. Current findings, thus, support low ANC use in Nigeria and highlight differences across national, rural, and urban settings.

Nationally, 69.3% of mothers received iron tablets or syrup, but disparities exist, with approximately one in five urban mothers and two in five rural mothers not receiving iron supplements. Our result for the national estimates (69.3%) compares with a Nigerian study (using the same dataset) [17] but is lower than in a previous study (using the NDHS 2013), where 90.8% received iron supplements [39]. These results indicate a decline in iron supplementation in Nigeria, with greater vulnerability for rural mothers. Additionally, the proportion of mothers who received medications for intestinal parasites (16%) was low but similar to previous reports of about 20% [39] and 16.7% [17] in the overall Nigerian context. Iron supplementation and treatment for intestinal parasites are essential components of ANC to reduce anaemia in pregnancy [1, 2]. Anaemia can contribute to numerous adverse outcomes, including preterm birth, low birth weight, severe postpartum haemorrhage, and foetal malformations [40, 41], underscoring the importance of the ANC components.

Following multivariable analyses, we identified several factors associated with ANC utilisation in Nigeria across rural and urban areas, revealing a complex interplay of intersectional dynamics. Often rooted in systemic inequalities, these dynamics can be better comprehended through the lens of social determinants of health [8, 9, 34]. The dynamics encompass various elements, including geographic location, regional disparities, healthcare system functionality, individual characteristics, socioeconomic status, and cultural influences, all intricately shaping access to and utilisation of healthcare services. Briefly, social determinants of health encapsulate how individuals live, learn, work, and play, shaped by the distribution of money, power, and resources [8]. These determinants profoundly influence a diverse spectrum of health outcomes and quality-of-life indicators [9]. Conversely, intersectionality elucidates how various systems of marginalisation and privilege intersect, intertwining to form unique experiences for individuals and communities [34]. This concept offers a framework for comprehending lived realities without reducing individuals to singular characteristics [34]. Therefore, promoting maternal and neonatal health outcomes in Nigeria necessitates a holistic approach that acknowledges and addresses the intertwined effects of social determinants and intersectional factors contributing to health inequities in the country. By embracing this comprehensive perspective, policymakers and healthcare stakeholders can devise more effective strategies to dismantle barriers to ANC use and advance the health and well-being of individuals and communities.

First, our study highlights stark geographic and regional disparities in ANC utilisation. Overall, rural-urban residences emerged as a significant predictor of ANC utilisation, underscoring the importance of the data disaggregation approach used in this study. Our analysis confirmed a consistent trend in previous studies [16, 24, 42, 43], with urban mothers having higher odds of ANC use than their rural counterparts, likely due to better infrastructure, more established health services, and greater access to social amenities. Regional differences were also pronounced, with southern regions generally exhibiting higher ANC utilisation than

northern regions, particularly the North-East. Many studies have reported regional differences or gaps between southern and northern Nigeria [5, 16, 44, 45]. These disparities may mirror the intersectionality of geographic location with other social determinants, such as security challenges, poor educational development, and low accessibility to healthcare services. We observed uniformly low ANC utilisation across the regions in rural northern Nigeria, indicating severe structural barriers. These findings emphasise the need for region-specific interventions that address systemic inequalities and unique local challenges, such as improving healthcare infrastructure, enhancing security, and increasing educational opportunities in northern Nigeria.

Second is the critical role of socioeconomic status and education in ANC utilisation. For example, mothers and their husbands/partners with higher education levels recorded greater odds of ANC contacts. This finding aligns with several national studies [16, 42, 43, 46], further underscoring the link between educational attainment and ANC utilisation. The consistency of this finding across Nigeria's national, rural, and urban settings, here and in our previous study [16], highlights the intersection of education and socioeconomic status as pivotal determinants of health. This premise aligns with the SDGs' emphasis on equal access to education for girls, indicating that education (SDG 4) can contribute to realising other SDGs, such as SDG 3.1, which aims to reduce the global maternal mortality ratio to less than 70 per 100,000 live births by 2030 [14]. Higher education enhances health literacy, empowers women to make informed health decisions, and increases access to resources necessary for regular ANC contacts. Consequently, policies promoting education, particularly for girls, are essential for improving ANC utilisation in Nigeria.

At the national level, our study reveals greater odds of ANC use among mothers from wealthy households. However, following data disaggregation, these differences only remained significant in rural settings, highlighting pronounced health equity gaps between the rich and the poor and socioeconomic disadvantage in accessing ANC services in rural Nigeria. The stark contrast in ANC utilisation between wealthy and low-income families in rural areas further illustrates the negative impacts of low socioeconomic circumstances. This finding may be consistent with our results of low ANC use among mothers not engaged in paid job in rural Nigeria. Previous studies have reported the vulnerability of women from low-income families, resulting in lesser use of maternal healthcare services in the overall Nigerian population [16, 47]. Current findings emphasise the need to prioritise rural mothers for improved ANC utilisation. Addressing these disparities through health insurance coverage and tackling socioeconomic inequalities is critical, particularly when viewed through intersectionality with geographic and socioeconomic factors.

Third, religious practices and ethnic identity were significantly associated with ANC utilisation patterns in rural settings. Notably, mothers who identified as Christians or Muslims had increased odds of ANC use compared to those who identified as Traditional or other religions, highlighting how religious affiliation intersects with other social determinants to influence health behaviours and access to services. In rural Nigeria, religious affiliation may interact with socioeconomic status, education, and community support structures. For example, Christian and Muslim communities might have better access to information about ANC services or more robust support networks encouraging healthcare utilisation. Conversely, those identifying with Traditional or other religions might face marginalisation or lack access to similar resources, potentially reflecting broader systemic inequalities.

Similarly, ethnic identity significantly predicted ANC utilisation, with the Hausa/Fulani recording disproportionately lower use, especially in urban settings. The Hausa/Fulani ethnic group, many of whom are nomads [48] and reside in more remote areas with limited healthcare infrastructure, can face additional intersecting barriers relating to lower socioeconomic

status, less education, and cultural practices that may not prioritise ANC. In addition, the pastoralists' nomadic lifestyle can exacerbate these barriers, for instance, through geographic isolation. Current findings underscore the importance of considering religious and ethnic identities in the context of other intersecting factors, such as geographic location, cultural norms, and socioeconomic conditions, when addressing ANC utilisation. Tackling these disparities requires culturally sensitive approaches that recognise and mitigate the barriers faced by mothers from less prevalent religious or ethnic groups with their unique characteristics. This approach must also address broader systemic inequalities to improve ANC use and overall maternal and neonatal health outcomes.

An essential consideration in addressing ANC underuse among different ethnic nationalities is adopting a culturally safe and responsive practice framework [49]. This framework ensures that health workers and professionals practice in ways that are respectful and supportive of the cultural identities and needs of diverse client populations [49]. It promotes an environment where clients feel understood, valued, and empowered, aiming to achieve equity and inclusion in service delivery [49]. Implementing this framework requires ongoing self-reflection, cultural competence training, and an active challenge to systemic inequities, and it should be an integral part of healthcare workers' training and education. Furthermore, gender norms and household power dynamics are crucial in ANC utilisation. Our study found that mothers with autonomy in healthcare decision-making were more likely to adhere to the recommended ANC contacts across all settings. Empowering women and promoting gender equity is, therefore, critical for improving ANC utilisation. This practice includes ensuring women have the knowledge, resources, and support needed to make informed healthcare decisions and addressing societal norms limiting women's autonomy. By incorporating these culturally sensitive and gender-responsive strategies, healthcare providers and policymakers can work towards eliminating barriers to ANC utilisation and achieving better maternal and neonatal health outcomes in Nigeria.

Fourth, the quality and accessibility of healthcare services are fundamental factors in ANC utilisation. Our study found that mothers faced significant barriers relating to inadequate or limited access to essential ANC components like iron supplements and medications for intestinal parasites. Addressing these systemic issues requires strengthening the healthcare system and ensuring well-equipped facilities with appropriately trained health professionals. It is also essential to maintain regular stocks of necessary medications and resources through efficient medicine supply chains [50]. This recommendation will help ensure timely access to safe, effective, and quality ANC components, thereby contributing to realising one of the major SDG 3 targets [14]. This observation is particularly crucial in rural and underserved regions where healthcare services may be less available or accessible.

Fifth, young mothers, especially teenagers, were less likely to utilise ANC services across all settings but more discernible in urban areas, highlighting the intersection of age and socioeconomic status. The pronounced disparity in urban areas suggests that younger mothers are more vulnerable. This distinction was less noticeable in rural areas, possibly due to more uniform challenges and similar levels of awareness and resources across different age groups. Young mothers often face multiple vulnerabilities, including limited education, economic constraints, and early marriage [5, 6], which can mutually reinforce their reduced access to ANC services. Current findings align with previous research [16], a recent investigation of age difference's association with 'non-use of ANC' in Nigeria [17], and a Nepalese study [51] indicating teenage mothers were less likely to be autonomous or able to exercise their sexual and reproductive rights. Targeted interventions, such as educational opportunities (e.g., scholarships), adolescent-friendly health services and appropriate sexual and reproductive health education

in schools and communities, can contribute to preventing early marriage or pregnancies among young girls and adolescents.

Mothers with four or higher birth orders were also less likely to achieve the recommended ANC contacts, possibly due to perceived experience from previous pregnancies or economic constraints. Current results are similar to those of earlier studies in Nigeria [16], Ghana [52], and other Sub-Saharan African countries [43]. Conversely, and consistent with our previous study [16], mothers with multiple deliveries (twins, etc) had higher odds of ANC use in rural areas, likely due to the increased perceived risk of complications. These findings suggest the need for targeted messaging and support for mothers based on their parity and unique health-care needs.

Lastly, internet and traditional media exposure emerged as significant factors associated with increased odds of ANC use across various settings in Nigeria. While nationally, the association was only marginally significant, the influence of Internet exposure suggests a potential avenue for disseminating information about ANC services, towards fostering awareness and encouraging health-seeking behaviours. This finding resonates with recent research highlighting the role of internet access in Caesarean Section utilisation, particularly in urban Nigeria [44]. The United Nations recognises unrestricted internet access as a fundamental human right [53, 54], underlining its importance. Furthermore, Internet connectivity and digital literacy are increasingly seen as 'super social determinants of health', capable of influencing other health determinants [55, 56]. Despite these advantages, most mothers in Nigeria do not use the Internet (90.6%), with rural areas showing even lower rates (97.3%). Ensuring equitable access to the Internet may be critical for enhancing healthcare services utilisation through access to information, telemedicine, and health promotional activities. This recommendation can be achieved through targeted investment in Internet facilities such as community Wi-Fi program, particularly in rural areas. The distinction in media influence between urban and rural areas is noteworthy: in urban settings, TV was associated with higher ANC use, while in rural areas, radio exposure played a crucial role. This finding reveals the importance of leveraging different media to reach diverse populations and encourage ANC utilisation.

## Strengths and limitations

There are notable strengths in our study. First, we used the recent WHO guidelines [1, 7] of ≥ 8 ANC as the recommended number of contacts, providing results with the potential for immediate policy implementation in Nigeria. To our knowledge, this is the first research using the new WHO guideline to examine ANC utilisation and associated factors at Nigeria's national, rural, and urban levels. Second, we performed analyses at overall, urban, and rural levels in Nigeria, enhancing the much-needed insights into the within and between population differences in the country. Our study, thus, provides evidence-based information for policies aimed at bridging equity gaps in maternal healthcare service use across geographic divides in Nigeria. This study used a nationally representative sample; therefore, findings are generalisable to the Nigerian population.

Some limitations, however, need to be considered in our study. Firstly, the study used data collected based on mothers' ability to recall ANC contacts. Recalling ≥ 8 ANC contacts may be challenging for some participants, which can contribute to over or under-reporting. However, in general, mothers tend to recall key childbirth events well, and we do not expect this limitation to alter the conclusion of our findings. Secondly, the study does not include all the variables that may be important for maternal healthcare service use. For instance, some cultural, economic, and contextual barriers to accessing ANC are not exclusively included in the NDHS 2018. Future studies may include those additional factors, including some qualitative

data, to explore ANC use in Nigeria further. Thirdly, the data utilised in this study are at least five years old and may not completely represent the current situation in the country. Nevertheless, the data remains the most recent of its kind in Nigeria. Being largely pioneering, our study lays the groundwork for further research in the country and other low-to-middle-income countries. Lastly, given that the data utilised in our study were based on a cross-sectional design, inferring causality between outcome and predictor variables is beyond the scope of this study.

## Conclusions

Our study reveals substantial disparities in ANC use, its components, and associated factors across Nigeria's national, rural, and urban settings. Nationally, ANC use is low, with only one-fifth of mothers achieving the recommended ≥8 contacts. Notable rural-urban disparities exist, with approximately 3.5 times higher ANC use in urban areas. The North-East region exhibits disproportionately lower ANC utilisation nationally and in urban settings, while the North-West shows the lowest use in rural areas. Our findings also highlight a concerning decline in receiving ANC components, such as iron supplementation, which particularly affects rural residents. Maternal and husband education levels, health insurance coverage, and maternal autonomy are significant predictors of ANC use across Nigeria. All ethnic groups showed higher ANC odds than the Hausa/Fulanis in urban areas, while only the Yorubas had greater odds in rural areas. Nationally, internet use was significant, with television being significant in urban areas and factors like maternal working status, wealth, birth type, religion, and radio listenership significant in rural settings.

Addressing the disparities observed in this study requires an intersectional approach that considers how multiple social determinants and identities, such as socioeconomic status and geographic location, intersect to impact healthcare access and utilisation. Tackling deep-seated social inequalities and mitigating barriers rooted in diverse cultural practices are essential for promoting culturally sensitive ANC services. Universal health insurance can minimise financial barriers, and enhancing internet access can facilitate information dissemination, health promotion, and telemedicine. Targeted media channels like radio and television can educate and disseminate information to marginalised communities. Empowering individuals and communities through educational opportunities can foster socioeconomic empowerment, maternal autonomy, and improved ANC use.

## Acknowledgments

We thank the DHS Program for granting us access to the NDHS dataset analysed in this study. We also thank the participating mothers who shared their time and information. Thanks to Edith Cowan University, Western Australia, for making research facilities available to the lead author.

## Author Contributions

**Conceptualization:** Emmanuel O. Adewuyi, Vishnu Khanal.

**Data curation:** Emmanuel O. Adewuyi.

**Formal analysis:** Emmanuel O. Adewuyi.

**Funding acquisition:** Emmanuel O. Adewuyi.

**Investigation:** Emmanuel O. Adewuyi.

**Methodology:** Emmanuel O. Adewuyi, Asa Auta, Mary I. Adewuyi, Yun Zhao, Vishnu Khanal.

**Project administration:** Emmanuel O. Adewuyi, Vishnu Khanal.

**Resources:** Emmanuel O. Adewuyi.

**Software:** Emmanuel O. Adewuyi.

**Supervision:** Emmanuel O. Adewuyi, Yun Zhao, Vishnu Khanal.

**Visualization:** Victory Olutuase.

**Writing – original draft:** Emmanuel O. Adewuyi, Asa Auta, Mary I. Adewuyi, Aaron Akpu Philip, Victory Olutuase, Yun Zhao, Vishnu Khanal.

**Writing – review & editing:** Emmanuel O. Adewuyi, Asa Auta, Mary I. Adewuyi, Aaron Akpu Philip, Victory Olutuase, Yun Zhao, Vishnu Khanal.

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
