## [Decision Letter · Decision Letter 0]

29 Apr 2024

PONE-D-24-05470Antenatal care utilisation in Nigeria: assessing disparities between rural and urban areas—a nationwide population-based studyPLOS ONE

Dear Dr. Adewuyi,

Thank you for submitting your manuscript to PLOS ONE. After careful consideration, we feel that it has merit but does not fully meet PLOS ONE’s publication criteria as it currently stands. Therefore, we invite you to submit a revised version of the manuscript that addresses the points raised during the review process.

We look forward to receiving your revised manuscript.

Kind regards,

José Antonio Ortega, Ph.D.

Academic Editor

PLOS ONE

Journal Requirements:

3. Please upload a copy of Supporting Information Figure/Table/etc. Supporting information which you refer to in your text on page 24 and 25.

**Additional Editor Comments:**

Two experts have reviewed the paper noting some limitations in reporting, that might require some changes in the analysis depending on the answer to the questions. Please pay attention, in particular, to the detailed suggestions of reviewer 2.

In particular, remember PLOS ONE policy regarding replication studies, that is of concern given the similarity to your previous study: If a submitted study replicates or is very similar to previous work, authors must provide a sound scientific rationale for the submitted work and clearly reference and discuss the existing literature.

Reviewers' comments:

Reviewer's Responses to Questions

**Comments to the Author**

1. Is the manuscript technically sound, and do the data support the conclusions?

Reviewer #1: Yes

Reviewer #2: Yes

2. Has the statistical analysis been performed appropriately and rigorously? 

Reviewer #1: Yes

Reviewer #2: Yes

3. Have the authors made all data underlying the findings in their manuscript fully available?

Reviewer #1: Yes

Reviewer #2: Yes

4. Is the manuscript presented in an intelligible fashion and written in standard English?

Reviewer #1: Yes

Reviewer #2: Yes

5. Review Comments to the Author

Reviewer #1: The main concept is fine. Data and research methodlogy including with statistics is acceptable. However, transition between pargraph and table is not well writen. It needs to check and revise a bit. Besides, discussion needs to related to their ressults. I hope they able to change and revise.

Reviewer #2: Overall impression

The paper will benefit from minor revisions. See comments below.

Abstract

1. Well-written

Consider a more actionable recommendation rather than repeating the phrase ‘vulnerable populations’

Introduction

2. The introduction could benefit from a more focused discussion on the specific research objectives and the rationale for examining rural-urban disparities in ANC utilization.

3. While it briefly mentions the previous study by the authors and the use of data disaggregation, a clearer explanation of why rural-urban disparities is particularly important in the Nigerian context would enhance the reader's understanding. Additionally, how the previous study differs from this except in the use of different datasets (ie 2013 vs. 2018).

Methods

4. Information on the study area, and contextual factors affecting ANC use is needed for a non-Nigerian reader to understand this. Consider shortening the data source to incorporate this.

5. We access the current data 158 for research purpose in June – August 2023. Should this be ‘accessed’?

6. Please clarify the role of the Children Recode dataset since it is focused on the record for children born in the preceding past 5 years. If this was used to access the mother’s information then make this explicit.

7. More details on the model building and comparison and the variables. What was used to compare models such as the Hosmer and Lemeshow tests and the area under the receiver operating characteristic curve (AUC).

8. Any need for sensitivity analysis? Why and why not

9. How did you choose variables to include the multivariable binary logistic regression analyses

10. Any information on cluster, strata

11. A DAG would be useful for this research and for model building as well as identifying variable types eg. Confounders, instrumental variables. Independent predictors, Mediators etc This will help decide what goes into the model and as what.

12. How was missingness handled eg. multiple imputation

Discussion and conclusion

13. It will be good to discuss using terms and concepts like intersectionality and social determinants of health which best describe the factors affecting ANC use

14. The discussion could benefit from a more nuanced focus on the implications for maternal and neonatal health outcomes in Nigeria

15. Consider making the conclusion shorter and removing proportions, including specific strategies for addressing the identified gaps. The below is by no means acceptable and quite non-specific!

‘Efforts to bridge the rural-urban, address regional disparities, and prioritise vulnerable populations are imperative for enhancing ANC utilisation and maternal and perinatal health outcomes in Nigeria.’

6. PLOS authors have the option to publish the peer review history of their article (what does this mean?). If published, this will include your full peer review and any attached files.

Reviewer #1: **Yes: **Yothin Sawangdee

Reviewer #2: No

---

## [Author Response · Author response to Decision Letter 0]

10 Jun 2024

Editor Comments

1. Two experts have reviewed the paper noting some limitations in reporting, that might require some changes in the analysis depending on the answer to the questions. Please pay attention, in particular, to the detailed suggestions of reviewer 2.

Response:

We thank the editor and reviewers for their feedback on our manuscript. We have carefully considered (and paid close attention to) all the comments and responded accordingly, making necessary corrections or providing rebuttals as appropriate. 

2. In particular, remember PLOS ONE policy regarding replication studies, that is of concern given the similarity to your previous study: If a submitted study replicates or is very similar to previous work, authors must provide a sound scientific rationale for the submitted work and clearly reference and discuss the existing literature.

Response:

We appreciate the editor's comment. We confirm no overlap between the present study and our previous publication (PMID: 29782511)[1]. Also, we did not utilise any information from the previous work except where appropriately cited. Below, we outline some differences between the two studies:

• Data used: Our previous publication (PMID: 29782511)[1] was based on data from the Nigeria Demographic and Health Survey (NDHS) 2013, while the current manuscript utilised entirely new and more recent data from the 2018 NDHS (the most current in the series of such surveys in Nigeria). Therefore, the present manuscript provides updated and more relevant evidence.

• Coverage: The previous publication solely addressed ANC underuse. Conversely, the current manuscript extends its coverage to include ANC utilisation and the quality of ANC received, thereby offering a more comprehensive assessment. The quality of ANC has been an issue and has rarely been prioritised in the literature from low-middle-income countries. Our study assessed ANC components and provided insights into differences across national, rural and urban settings in Nigeria.

• Definition: The previous publication adhered to the Focused ANC model, which typically requires a minimum of four ANC contacts. However, the current manuscript adopts the new WHO recommendation of eight or more contacts [2, 3], reflecting alignment with updated global health guidelines.

(Web link to reports on WHO's new guideline on ANC: 

https://www.who.int/news/item/07-11-2016-new-guidelines-on-antenatal-care-for-a-positive-pregnancy-experience)

Thus, the current study builds upon previous research by utilising more recent data and expanding the scope of analysis to include both ANC utilisation and quality. The adoption of updated WHO guidelines underscores the relevance and rigour of our findings in informing maternal healthcare policies and practices in Nigeria with a focus on rural-urban differences. We have updated our manuscript to reflect the differences as appropriate.

Reviewers' comments

Reviewer #1: 

The main concept is fine. Data and research methodlogy including with statistics is acceptable. However, transition between pargraph and table is not well written. It needs to check and revise a bit. Besides, discussion needs to related to their ressults. I hope they able to change and revise.

Response:

We thank the reviewer for the insightful, helpful, and constructive feedback. We have carried out a thorough revision, improving on transition between paragraphs and the presentation of our Tables. Also, we have improved or even rewritten some parts of the discussion, ensuring it focuses on findings.

Specifically, we have made the following revisions, which can be seen in the tracked version of our revised manuscript:

• We improved the readability and presentation of the abstract section, ensuring the recommendations are more actionable.

• We improved the presentation and flow of the introduction section, ensuring that transitions between paragraphs are seamless and connect well.

• We improved the presentation of our Tables to make them compact and easy to read. Our tables are relatively large but have been presented in ways that make them easy to follow.

• We carried out a thorough proofreading of our manuscript and made corrections as necessary.

Reviewer #2: 

Overall impression: The paper will benefit from minor revisions. 

Abstract

1. Well-written. Consider a more actionable recommendation rather than repeating the phrase ‘vulnerable populations’

Response:

We appreciate the reviewer for the insightful, helpful, and constructive feedback. 

We have revised the abstract section for clarity. Importantly, we provide more actionable recommendations (with consideration for word count limit) and avoid repetition of words or phrases, including ‘vulnerable populations’.

The conclusion (with brief recommendations) in the abstract section now reads: 

‘Our study reveals significant disparities in ANC utilisation and components across Nigeria, with rural residents, particularly in northern regions, socioeconomically disadvantaged and teenage mothers facing notable challenges. A multifaceted approach prioritising the interplay of intersectional factors like geography, socioeconomic status, education, religion, ethnicity, and gender dynamics is essential. Key strategies should include targeted interventions to promote educational opportunities, expand health insurance coverage, leverage internet and context-specific media, and foster socioeconomic empowerment, prioritising underserved populations’ (lines 57 – 63).

Introduction

2. The introduction could benefit from a more focused discussion on the specific research objectives and the rationale for examining rural-urban disparities in ANC utilization.

Response:

We appreciate the merit of this comment and have rewritten several aspects of the introduction section. Specifically, we made our introduction more focused and highlighted specific research objectives, backed by the rationale for focusing on rural-urban disparities in ANC utilisation. Below, we provide some excerpts:

• Specific objectives:

‘The present study's objectives, thus, are to assess ANC utilisation of eight or more contacts (following the new WHO guidelines) and evaluate its variation across national, rural, and urban settings in Nigeria. Secondly, we aim to examine the receipt of ANC components and their differences across these settings. Additionally, we investigate geographic, demographic, socioeconomic, and healthcare-related factors associated with ANC use and assess how they differ across national, rural, and urban settings’ (lines 149 – 155).

• Rationale for rural-urban assessment:

‘However, several available studies are limited by their generalised approach, primarily focusing on national estimates using pooled datasets, which may inadvertently mask differences between and within population groups [1]. There is a growing appreciation of the importance of using high-quality, disaggregated data studies as an evidence-based approach to address access, survival, and equity disparities across socioeconomic and geographic divides [4-6]. This type of data disaggregation aligns with the WHO’s framework for monitoring progress towards Universal Health Coverage, which emphasises the need to disaggregate all socioeconomic and demographic strata measures to better assess equity in intervention coverage, among other factors [7]’ (lines 115 – 122)

3. While it briefly mentions the previous study by the authors and the use of data disaggregation, a clearer explanation of why rural-urban disparities is particularly important in the Nigerian context would enhance the reader's understanding. Additionally, how the previous study differs from this except in the use of different datasets (ie 2013 vs. 2018).

Response: We have provided additional information and refined our approach in the introduction to clarify our study’s focus on data disaggregation by the rural-urban divide in Nigeria. Also, we highlight how the present study differs from our previous publication in this area.

• Data disaggregation approach:

‘Importantly, in its newly released 2024 ‘Operational Framework for Monitoring Social Determinants of Health Equity’ [8], the WHO emphasises that data disaggregation is crucial for monitoring social determinants of health as it enables the identification and analysis of health inequities across different population subgroups. For example, by breaking down data by factors such as geographic location (rural-urban), demographic (age, gender), socioeconomic, and ethnicity, it is possible to uncover disparities and target interventions more effectively [8]. This detailed analysis approach supports the goal of the 2030 Agenda for Sustainable Development to ‘leave no one behind’ by tracking progress across various Sustainable Development Goals (SDGs) and ensuring equitable access to health-promoting resources and services [8].

Specifically, Nigeria's diverse demographics and healthcare infrastructure underscore the need for understanding ANC utilisation across various subpopulations [9]. The country's notable geographic, demographic and socioeconomic disparities can exacerbate existing health inequities and have implications for policymaking and resource allocation efforts to improve maternal healthcare services. Identifying areas where resources and interventions are most needed can, for example, enable tailored strategies to bridge the gap in ANC utilisation and improve maternal and neonatal health outcomes nationwide’ (lines 122 – 137).

• Comparison with our previous publication:

‘Consistent with this premise, our previous study used data disaggregation to gain insight into Nigeria's underutilisation of ANC services, focusing on differences between rural and urban residences [1]. Given the importance of this subject and the new WHO recommendations, we aim to build upon the study by investigating ANC service utilisation and quality (i.e., receipt of ANC components) in Nigeria using the latest nationally representative demographic and health survey data [9]. 

While our prior study [1] adhered to the focused ANC model, requiring a minimum of four ANC contacts, the current study adopts the new WHO recommendation of eight or more contacts [2, 3, 10], reflecting alignment with the updated global health guidelines. In addition, we expanded our investigation to examine disparities in receiving essential ANC components between rural and urban areas following the recommended data disaggregation approach. These components, including iron supplementation, tetanus vaccinations, medicines for intestinal parasites, blood pressure checks, and various laboratory tests [2, 3], are crucial for promoting maternal and neonatal health’ (lines 137 – 149).

Methods

4. Information on the study area, and contextual factors affecting ANC use is needed for a non-Nigerian reader to understand this. Consider shortening the data source to incorporate this.

Response: We have provided a brief sub-section for the study setting, introducing the country and setting the tone for understanding our study. This sub-section reads:

‘Nigeria is Africa's most populous country, with over 220 million people, nearly half (46.48%) of whom reside in rural areas. The country comprises over 374 ethnic groups and languages, organised into 36 states and the Federal Capital Territory, with six geopolitical zones (regions). Further administrative subdivisions include local government areas (LGAs) within states and wards, also known as enumeration areas (EAs), within LGAs. Approximately 63% of Nigerians live in multidimensional poverty [11], highlighting socioeconomic disparity. Healthcare provision involves both public and private sectors, with public health services shared among the three tiers of government: primary (LGAs), secondary (states), and tertiary level of care (federal). In 2017, a notable shift occurred in ANC practices following the adoption of the 2016 WHO ANC guideline [2, 3], increasing ANC contacts from a minimum of four to eight in Nigeria [10]. Nigeria struggles with poor population health outcomes. However, national statistics can mask the profound disparities across geographic and socioeconomic divides [12], hence, the rural-urban data disaggregation approach used in the current study’ (lines 168 – 179).

5. We access the current data 158 for research purpose in June – August 2023. Should this be ‘accessed’?

Response: Thank you for picking the grammar error; we have corrected it to ‘accessed’ (line 201).

6. Please clarify the role of the Children Recode dataset since it is focused on the record for children born in the preceding past 5 years. If this was used to access the mother’s information then make this explicit.

Response: We have provided the information requested, which reads: 

‘The KR dataset contains individual records for each child born to the interviewed women within the five years before the survey. It provides extensive details regarding the child's pregnancy, postnatal care, immunisation, general health status, and corresponding data for their mothers. We utilise the data available in this dataset to access the mother's information. We received approved access to use the data, and no further ethical clearance was required to conduct this study’ (lines 202 – 206).

Link to the DHS website for more information: https://dhsprogram.com/data/Dataset-Types.cfm

7. More details on the model building and comparison and the variables. What was used to compare models such as the Hosmer and Lemeshow tests and the area under the receiver operating characteristic curve (AUC).

Response: The survey producing the data for our analysis is based on a complex sample design with survey weights, clustering, and stratification, which we reckon can complicate model fit testing in logistic regression analysis. We did not use Hosmer-Lemeshow or AUC tests in the present study. The Hosmer-Lemeshow test is one of the most widely applied tests in practical applications, assuming a simple random sample. However, this assumption may not be valid for survey data based on complex sample designs. Notably, the test is unavailable in the CS of SPSS (to our knowledge). Given the nature of our study and to avoid complexities, we followed approaches in previous studies where these tests were not used [1, 13-20]. To address concerns about likely statistical errors, we note as follows:

‘To minimise potential statistical errors, we meticulously double-checked our analysis process to ensure all variables meeting the inclusion criteria were included in our models. Additionally, we rigorously validated the effectiveness of our backward elimination modelling by testing the final parsimonious models against potential confounding variables and factors previously reported to be associated with ANC’ (lines 265 – 269).

8. Any need for sensitivity analysis? Why and why not

Response: Sensitivity analysis in logistic regression for survey data with stratified sampling design may be complicated, with the potential for biased results. While we cannot rule it out, to the best of our knowledge, most previous studies in this area of research do not perform sensitivity analysis [1, 13-20]. 

9. How did you choose variables to include the multivariable binary logistic regression analyses

Response: we provided information on our multivariable model building, which reads as follows:

‘In our model-building process, we only included variables that showed statistical significance (p < 0.05 in the Chi-Square test) in the initial model of the multivariable regression analyses’ (lines 261 – 263).

So, variables significantly associated with ANC use at p < 0.05 (in the univariate analysis) were selected for inclusion in our multivariable logistic regression modelling. 

10. Any information on cluster, strata

Response: Yes. We noted in our manuscript as follows:

‘The NDHS 2018, the sixth edition in its series, used a two-stage stratified cluster sampling method involving about 42,000 households and 1,400 clusters [9]’ (lines 189 – 190)’. 

Also, we mentioned in the statistical analysis sub-section that:

‘In all analyses, we used the Complex Sample analysis method, considering the stratified design and the samplin

---

## [Decision Letter · Decision Letter 1]

1 Jul 2024

Antenatal care utilisation and receipt of its components in Nigeria: assessing disparities between rural and urban areas—a nationwide population-based study

PONE-D-24-05470R1

Dear Dr. Adewuyi,

We’re pleased to inform you that your manuscript has been judged scientifically suitable for publication and will be formally accepted for publication once it meets all outstanding technical requirements.

Kind regards,

José Antonio Ortega, Ph.D.

Academic Editor

PLOS ONE

Additional Editor Comments (optional):

The two previous reviewers were reached for comment. Only reviewer one was available recommending acceptance. As the academic editor I have inspected whether my own comments and those of reviewer 2 were satisfactorily addressed and they are. The difference with the previous research is now clear and the comments raised in the previous version have indeed been taken care of. 

Reviewers' comments:

Reviewer's Responses to Questions

**Comments to the Author**

1. If the authors have adequately addressed your comments raised in a previous round of review and you feel that this manuscript is now acceptable for publication, you may indicate that here to bypass the “Comments to the Author” section, enter your conflict of interest statement in the “Confidential to Editor” section, and submit your "Accept" recommendation.

Reviewer #1: All comments have been addressed

2. Is the manuscript technically sound, and do the data support the conclusions?

Reviewer #1: Yes

3. Has the statistical analysis been performed appropriately and rigorously? 

Reviewer #1: Yes

4. Have the authors made all data underlying the findings in their manuscript fully available?

Reviewer #1: Yes

5. Is the manuscript presented in an intelligible fashion and written in standard English?

Reviewer #1: Yes

6. Review Comments to the Author

Reviewer #1: The main revision is acceptable. It is fine and able to answer research objectives. Importantly, the transition between paragraph and tables, graph is acceptable. Discussion and citations is fine. So, I would suggest to accept.

7. PLOS authors have the option to publish the peer review history of their article (what does this mean?). If published, this will include your full peer review and any attached files.

Reviewer #1: **Yes: **Yothin - Sawangdee

---

## [Editor Report · Acceptance letter]

4 Jul 2024

PONE-D-24-05470R1 

PLOS ONE

Dear Dr. Adewuyi, 

I'm pleased to inform you that your manuscript has been deemed suitable for publication in PLOS ONE. Congratulations! Your manuscript is now being handed over to our production team.

Kind regards, 

on behalf of

Dr. José Antonio Ortega 

Academic Editor

PLOS ONE